

# African mineral dust sources: a combined analysis based on 3D dust aerosols distributions, winds and surface parameters

Sophie Vandenbussche[1] and Martine De Mazière[1]

[1]Royal Belgian Institute for Space Aeronomy, 3 Avenue Circulaire, 1180 Brussels, Belgium

*Correspondence to:* Sophie Vandenbussche (sophie.vandenbussche@aeronomie.be)

**Abstract.** Mineral dust aerosols are a key player in the climate system. Their emissions are not yet characterised enough to ensure their good representation in climate models. The work presented here aims at a better characterisation of dust sources by a new analysis method. We use the three-dimensional dust aerosols distribution from the Infrared Atmospheric Sounding Interferometer (IASI), obtained with the Mineral Aerosols Profiling from Infrared Radiances (MAPIR) algorithm. The availability of vertical information on top of total column information allows to better separate emissions from transport. However, the presence of dust at the surface could also be due to low altitude transport, or to deposition processes. Therefore, to strengthen the analysis, we have completed it with an analysis of wind speed and surface state parameters (land cover, vegetation, moisture). For the more complex case of the Sahel, we have also analysed the soil type and the wind direction patterns. Our analysis highlights the well-known Saharan hot-spots, but also a less well-known significant emission place west of the Bodélé depression. The study of Sahel dust sources is a new feature for satellite-based analyses. Our results are coherent with those drawn from local ground-based measurements, allowing to extend our analysis to the entire Sahel area with confidence. We also provide a morning versus evening comparison, helping to distinguish the different emission mechanisms in play, and a small year-to-year variation analysis.

## 1 Introduction

Mineral dust aerosols are one of the most prominent types of tropospheric aerosols, accounting for about 35% of the total aerosol mass with diameter smaller than 10 $\mu m$ (Boucher et al., 2013). These aerosols are mainly observed in the so-called "dust belt" in the North Tropical area, taking their origin in the Saharan, Middle East and Asian deserts. They are mineral particles uplifted from bare soils by winds, which may stay in suspension for a few days, even weeks, allowing their transport along thousands of kilometres, for example from the Sahara across the Atlantic ocean all the way to America. Dust aerosols are most often found at altitudes below 7 $km$ due to their size (Tsamalis et al., 2013), contrary to mineral ash particles that may be found up to 20 $km$ altitude (Maes et al., 2016).

Mineral aerosols impact the Earth's energy balance along the whole solar (short-wave) and terrestrial (long-wave) spectral ranges, due to the large particle size and the existence of significant absorption features in both visible and Thermal Infrared (TIR) ranges (e.g. Sokolik et al., 1998). More precisely, uplifted mineral dust particles absorb and scatter the solar, Earth and atmosphere-emitted light, and they emit thermal radiation themselves. These so-called direct effects may result in changes





of the surface and atmospheric temperatures, which are then called semi-indirect effects (Choobari et al., 2014). The sign and extent of such changes vary with surface parameters, aerosol properties and vertical distribution (Boucher et al., 2013) and may lead to changes in wind patterns (Choobari et al., 2014). In addition, mineral aerosols act as ice condensation nuclei, modifying the number and size of droplets and therefore affecting the cloud lifetime and the amount of rain (Choobari et al., 2014). These

effects are called the indirect effects. Mineral aerosols also play a role in the transport of soil nutrients to the oceans and to the Amazon forest (e.g. Rizzolo et al., 2017).

Mineral dust is a known hazard for human health, causing respiratory, cardiovascular disorders and infectious diseases, the latter due to the presence of micro-organisms on dust particles (Zhang et al., 2016; Middleton, 2017). In addition, the suspended particles may reveal problematic for several forms of communication, including radio waves and microwaves and can cause

dangers for air transportation as much as does volcanic ash (Middleton, 2017).

Although mineral aerosols are in essence natural particles, emitted through natural mechanisms, part of their emissions and therefore part of their atmospheric burden is linked to human activities. Indeed, these activities disturb the soil and modify the vegetation cover, often leading to more erodibility of the surface. Dust emissions are also indirectly impacted by climate changes in general, which impact the soil temperature and humidity, and the wind patterns (Choobari et al., 2014). Ginoux

et al. (2012) have used MODIS data to extract mineral dust source information with emphasis on the natural or anthropogenic origin of the emissions. For example, they showed that most of the dust emissions in the Sahel have a link to human activities, while most of the emissions in the Saharan desert are purely natural.

Significant uncertainties remain as to the global effect of mineral dust aerosols on the climate (Boucher et al., 2013). The goal of this work is to provide additional information about mineral dust sources, to complete the current knowledge. For that

purpose, we use a newly obtained three-dimensional mineral dust aerosols distribution, combined with surface wind fields and information about land cover, vegetation and the surface state. In this manuscript, we will describe the dust emission mechanisms, explain the added value of using satellite data from the Infrared Atmospheric Sounding Interferometer (IASI), describe how the new 3D IASI data is used, and develop a complex analysis method to distinguish between transported dust aerosol, dust deposition and local sources. We will then analyse the results over north Africa in terms of global source areas,

seasonal cycle, diurnal variations and long-term evolution. A special emphasis will be given to the Sahel, which is a more complex place due to the succession of a dry and a wet season. In Appendix A, the MAPIR algorithm used to retrieved the 3D dust distribution from IASI data is detailed.

## 1.1 Mineral dust emission mechanisms

Mineral dust emissions occur by the wind erosion of bare surfaces. Three different mechanisms have been highlighted for these

emissions, depending on the particle size and soil composition (e.g., Gherboudj et al., 2016; Marticoréna, 2014): creeping of the largest particles, saltation for the middle size particles (diameter between 70 and 500 $\mu m$) and suspension of fine particles. Creeping occurs for particles that can not be uplifted (too heavy and/or low wind); they are rolled over the surface and may induce emission of smaller particles upon impact. Saltation is considered the most efficient emission mechanism (50 to 90% (Gherboudj et al., 2016)) and consists of subsequent short jumps along the surface which induce disaggregation in smaller





particles or emission of other particles, a process called sandblasting. Suspension occurs when fine particles are directly uplifted by strong winds. However, this mechanism is reported to account for only about 1% of the total dust emissions (Gherboudj et al., 2016).

Under a certain wind threshold, called the wind erosion threshold (Marticoréna, 2014) or the threshold friction velocity (Gherboudj et al., 2016), the particles can't be entrained. This threshold depends on the surface properties, including surface roughness, particle size and composition, humidity. That threshold is usually reported to be 5 to 6 *m/s* (Marsham et al. (2013) and Marticoréna (2014)), while the most intense emissions occur when the surface wind exceeds 8 to 10 *m/s* (Kocha et al. (2013) and Marticoréna (2014)).

There are two major mechanisms known to be responsible for intense surface winds: convective events and the turbulent mixing of the nocturnal low-level jets (LLJ) (e.g., Schepanski et al., 2009; Knippertz and Todd, 2012; Heinold et al., 2013; Allen et al., 2013). LLJs are strong horizontal winds in the lowest atmospheric layers (usually at the top of the boundary layer), most often active during the night. After sunrise, the onset of solar heating causes their turbulent breakdown resulting in high surface wind speeds during the mid-morning. This mechanism has been shown to be a significant player in Saharan dust emissions with a significant seasonal cycle (e.g., Fiedler et al., 2013). They are most active during the winter at the Bodélé depression and other source areas in the centre and east of the Sahara, and during the summer in West Sahara, in the Sahel and north of Libya (Fiedler et al., 2013). Dust emitting convective events are of many different origins and occur at different scales, from the micro scale to the synoptic scale (Knippertz and Todd, 2012; Knippertz, 2014). For example, at the largest scale, one finds cyclones forming south of the Atlas mountains during spring then moving eastward along the Mediterranean coast (Schepanski et al., 2009; Knippertz and Todd, 2012). A second example are the so-called "haboobs", mesoscale (up to 500 *km*) dust emissions due to the down-burst of cold and humid air related to deep moist convection that happen during the summer late afternoons and nights (Schepanski et al., 2009; Knippertz and Todd, 2012; Heinold et al., 2013). They occur frequently in the Sahel and southern Sahara (Knippertz, 2014), and can raise substantial walls of dust (Allen et al., 2013). At the very small scale, there are the "dust devils" (compact rotating dry dust plumes of about 10 *m* diameter) and larger non-rotating dusty plumes (100 *m* diameter); occurring at the time scale of 10 minutes to an hour (Knippertz and Todd, 2012; Knippertz, 2014). They occur under dry conditions and intense surface heating, and may reach 1 to 2 *km* altitude (Allen et al., 2013). They are extremely difficult to detect with satellite data, due to their small scale in both time and space.

Dust emissions are known to occur with a diurnal cycle (e.g., Schepanski et al., 2009; Heinold et al., 2013; Kocha et al., 2013; Banks et al., 2014). There are two maxima: a sharp one at about 9h local time, usually attributed to the LLJs effect, and a broader maximum during the late afternoon and night, mostly due to convective events (Heinold et al., 2013). Each source area is susceptible to experience only one or both of these mechanisms, with possible seasonal variations linked to wind seasonal changes.

## 1.2 Short review of African dust source studies using satellite measurements

African dust sources are studied using in situ measurements, satellite measurements or models. There is a significant variation in the results of the different studies, linked to the strengths and weaknesses of each measurement system, of each analysis



method, and to the time of the measurements. This manuscript is not intended as a review of the field, therefore we will provide only some examples of satellite-based dust source studies. Prospero et al. (2002) have studied dust sources in Africa and the Middle East based on TOMS Aerosol Absorption Index for years 1981 to 1986. Schepanski et al. (2012) have compared dust source analyses with the same method but based on measurements by the satellite instruments SEVIRI, MODIS (Deep

Blue) and OMI, showing significant differences among each other. Ashpole and Washington (2013) have developed a way of automatically tracking dust plumes observed by SEVIRI to identify their source region. Recently, Todd and Cavazos-Guerra (2016) used CALIOP data to add altitude information to the analysis, and Parajuli and Yang (2017) have jointly analysed MODIS Deep-Blue Aerosol Optical Depth (AOD) with wind and surface fields for the Bodélé area. This shows a growing interest in the field. However significant discrepancies remain between analyses based on different data and more research is

necessary.

### 1.3   Added value of IASI measurements for dust source studies

The Infrared Atmospheric Sounding Interferometer (IASI) was developed by the Centre National d'Etudes Spatiales (CNES) and is exploited by the European Organisation for the Exploitation of Meteorological Satellites (EUMETSAT). IASI is flying onboard the Metop satellite series on mid-morning sun-synchronous polar orbits (9h30 local solar time at equator). IASI is

a Fourier-Transform Michelson interferometer measuring at nadir the Earth and atmosphere emissions and solar backscatter between 645 and 2760 $cm^{-1}$. Its resolution is 0.5 $cm^{-1}$ after apodization, and the radiometric noise is evaluated to about 0.2 $K$ in the TIR atmospheric window (Clerbaux et al., 2009). IASI has a swath width of 2200 $km$ corresponding to a maximum viewing angle of 48.3° on both sides off nadir. Each across-track scan is composed of 30 elementary fields of view, each composed of 4 instantaneous fields of view of 12 $km$ diameter at sub-satellite point, growing to an ellipse of 39 by 20 $km$ at the edge of the

scan line.

IASI provides data time series adapted for climate studies, with measurements since 2007 and planned in the future until at least 2022, using three successive identical instruments flying on-board Metop-A, B and C. The long-term future is also ensured by the second generation instruments in preparation, with higher resolution and better signal-to-noise ratio.

The local solar time of IASI measurements is particularly interesting for studying African dust sources. Indeed, the mid-

morning measurements occur shortly after the breaking of the nocturnal LLJs responsible for a significant part of the dust uplifting. The evening measurements occur a short time after the second maximum of the diurnal dust emissions (Heinold et al., 2013). Using IASI therefore allows to gain information right after the two diurnal peaks of dust emissions and therefore probably tends to provide higher AODs close to sources, but also a better representation of the source areas, while instrument measuring at mid-afternoon (as MODIS on-board Aqua, CALIOP, OMI) might observe mainly dust downwind from sources

(Schepanski et al., 2012).

Almost all dust source studies undertaken up to now using satellite data are based on total column measurements, containing all aerosol species in one single AOD with no vertical information. This requires the ability to distinguish local emissions from advection and to isolate dust aerosols, which is usually done using Angström exponents (as particle size proxy). However, big particles may originate in transport from distant sources rather than being locally produced (Allen et al., 2013). In addition, in





case of night-time measurements, the Angström exponent method does not work because it requires the use of sun-photometers. IASI offers the advantage that the TIR atmospheric window is sensitive only to coarse mode aerosols, including mineral aerosols and sea spray aerosols. The latter do not have absorption features in the TIR, therefore only their scattering effect could be seen, remaining rather low. In addition, sea spray aerosols are not expected above dust source areas. On the other

hand, mineral aerosols like dust or volcanic ash have significant absorption features in the TIR. As a consequence, IASI measurements allow to derive a mineral aerosol product with no influence from other aerosol types present over source areas. The SEVIRI instrument, onboard Meteosat second generation in a geostationary orbit, also provides measurements in the TIR, but only in a limited number of low resolution channels, allowing to retrieve a dust presence index but not AOD nor vertical information (e..g. Schepanski et al., 2007). CALIOP delivers aerosol-typed high resolution vertical profile data, used in the

recent study by Todd and Cavazos-Guerra (2016). However, the ground coverage of CALIOP measurements is extremely poor, likely missing many dust events and therefore misrepresenting the global picture. IASI on the other hand offers almost global coverage twice a day for each flying instrument. Recently, we have developed the Mineral Aerosols Profiling from Infrared Radiances (MAPIR) algorithm for the retrieval of dust vertical profiles using IASI (Vandenbussche et al., 2013). It does not provide as good a vertical resolution as CALIOP does but it allows a significantly better ground coverage. This algorithm is

described in Appendix A.

To summarize: the strengths of our IASI data chosen for this study are that they provide 10 years of continuous consistent measurements (and more to come), global Earth coverage twice per day at very interesting times regarding the dust emissions, targeting only mineral aerosols including their vertical distribution.

## 2   Mineral dust source analysis method

### 2.1   10 years of IASI 3D dust distribution

The dust sources analysis method that we propose here is based mainly on the use of the IASI MAPIR 3D aeolian dust distributions, and additional data characterizing the surface, the vegetation and the winds. The MAPIR algorithm provides vertical profiles of mineral dust aerosols for each cloud-free IASI spectrum. Those profiles are provided as particle concentrations every kilometre from 1 to 6 $km$ altitude. Remember that these concentrations are not fully independent from each other, nor from the

a priori concentrations (see next section 2.1.1 for the details). MAPIR data is used for its ability to distinguish the presence of dust not only in the total column but also specifically in the atmospheric layer closest to the surface (which we refer to as "surface layer" in this manuscript). Considering that the vertical sampling of MAPIR is 1 $km$, this first retrieval point above the surface may be a few hundred meters above the surface. However, intense dust events have a non-negligible vertical extension (2 to 5 $km$ (Allen et al., 2013)) which is compatible with this limitation.

The application of the product to the study of desert dust sources as undertaken in this work does not require validation of the AOD nor of the full 3D distribution, but it requires a validation of the surface dust detection, which will be done in section 2.1.2, after describing the specific quality control (in section 2.1.1) and surface dust detection method.



### 2.1.1 Tailored quality control of MAPIR data

For our application, we consider only the MAPIR retrievals above land surfaces, as obviously oceans cannot be dust aerosols sources. For the application of the data to dust sources studies, we rely on the differentiation of the presence of dust in the total column or in the surface layer. It is therefore important to consider only MAPIR data for which there was enough sensitivity

to the surface layer so that the retrieved surface dust concentration is physically dependent on the real surface concentration at the time of the IASI measurement, and not only on the a priori. An advantage of the optimal estimation retrieval method used in MAPIR (see Appendix A) is that it provides Averaging Kernels (AKs), that quantify the sensitivity of a retrieval to the different retrieval altitudes and its dependence on the a priori concentrations. AKs are square matrices of the size of the state vector (for MAPIR: Ts and the dust concentration at the 6 retrieval levels). A value of 1 for a diagonal element means

that the corresponding retrieved parameter is equal to the true parameter; a lower value means that the a priori contributes to the retrieved value to some extent (the difference between 1 and the AK value). The non-diagonal elements report the cross-sensitivity of the retrieved parameters. The trace of the AK (sum of the diagonal elements) reports the number of Degrees of Freedom (DOFs) in the retrieval, or independent pieces of information originating from the measurement. For mineral aerosols retrievals from IASI TIR data, the DOFs in the vertical profiles of aerosols are reported to be 2 in the best situations

(Vandenbussche et al., 2013; Cuesta et al., 2015). That means that even though the vertical grid of the MAPIR retrieval has 1 $km$ intervals, the retrieved profile's vertical resolution is coarser than that, and the retrieved concentration at each level depends on the concentrations at the adjacent levels and on the a priori. The DOFs are distributed differently for different atmospheric/surface conditions, and it is therefore not possible to design a retrieval that would for example consider only two retrieval altitudes, each of them being totally independent. A common mistake for profile retrievals with 2 DOFs is to conclude

that a profile may not be retrieved, but only a total column (or AOD) and a mean altitude. This is true only if those 2 parameters are independent, which is not the case for the TIR mineral dust aerosols retrievals (Vandenbussche et al., 2013).

Coming back to the dust sources studies and the need for sensitivity to the surface layer, one must inspect the AK diagonal value for the surface layer parameter. As mentioned above, the total DOF for the full 6 points aerosol profile retrieval is 2 in the best cases. That means on average about 0.33 DOF for each retrieval point. However, the TIR sensitivity to the lowest

layers is often smaller than for higher layers. Indeed, mineral aerosols in the TIR absorb, emit and scatter light. The first two effects cancel each other when the temperature of the aerosols equals the temperature of the Earth surface. In consequence, for atmospheric layers with a temperature close to that of the surface, most of the sensitivity to the mineral aerosols comes from the scattering component of their extinction, which is the smallest contribution (single scattering albedo is about 0.6 at TIR wavelengths (Vandenbussche et al., 2013)). Therefore, we have set a threshold at 0.25 DOF minimum. Considering that

this threshold is far from 1, it means that the retrieved dust concentration in that layer depends also on the dust present in the layer(s) above. That means that in any case, we should not consider the absolute value of the retrieved concentration in a single layer. However if using a high enough concentration threshold we obtain a good indicator of the presence of dust in that layer.



### 2.1.2 Surface dust detection with MAPIR

The selection of the threshold in surface dust concentration for considering a scene to be dusty was done together with a validation of the MAPIR surface dust detection. Indeed, as just explained, the retrieved concentration in a single layer should not be considered quantitatively. Therefore establishing a threshold for the dust presence can not be done by evaluating physically which amount of dust aerosols makes a scene dusty. A full validation of the method is impossible for the reasons explained hereunder within the comparison analysis.

Comparisons are done with the extinction profiles from the Cloud-Aerosol Lidar with Orthogonal Polarization (CALIOP) onboard the CALIPSO platform (Winker et al., 2009) on a sun-synchronous afternoon orbit. We use the extinction at 532 *nm*, labelled as dust or polluted dust, from the newest version 4.1 of the data. We analyse IASI-CALIOP coincidences along the year 2015 in North Africa.

IASI measures at local solar times of 9h30 and 21h30 for the centre of the track. As the width is of 2200km, the local solar time at the sides of the track is of about 8h50 and 10h10 in the morning (and similarly in the evening) at the equator. CALIOP measures at a local solar time of 01h30 and 13h30. Therefore, to compare the CALIOP and IASI measurements closest in time, a maximum time difference of 5 hours is selected. The maximum spatial difference has been set to 50 *km* and all colocations are used in the statistics (not only the closest match in space).

We compared surface dust detections with MAPIR and with CALIOP, for different detection thresholds. For MAPIR, the surface dust detection is done after the specific quality filtering, and considering only the dust concentration in the first retrieval layer above the pixel mean surface altitude. For CALIOP, the dust extinction is integrated along a 1 *km* layer above the surface to obtain a "surface dust AOD". We have tested a number of different pairs of MAPIR concentration and CALIOP AOD thresholds and we retained the pair that leads to the best comparison statistics. This might seem a biased approach but it is simply due to the fact that the single layer concentrations retrieved with MAPIR may not be used quantitatively. The as such empirically selected thresholds are 50 *particles/cm³* (10 *μm* AOD of 0.2 in a 1 *km* thick layer) for MAPIR surface concentration and a 1 *km* layer AOD of 0.05 at 532 *nm* for the CALIOP integrated surface layer. Both seem to be the minimum thresholds for reliable dust detection. The latter is clearly lower for CALIOP because its sensitivity is much higher.

We have analysed the surface dust detection statistics with histograms (detection by only one sensor, both or none) for different periods (the whole year or each season separately). Due to the diurnal cycle of dust emissions (see section 1.3), it is expected that IASI detections, occurring right after the emission peaks, are more numerous than CALIOP detections, occurring during the emission minima. This is especially expected for the IASI morning versus CALIOP afternoon measurements: the morning emission peak is sharp in time, and the afternoon minimum is the lowest minimum of the day. In addition, if transport in the surface layer occurs, considering up to 5 hours time difference, a wind speed of only 5 *km/h* (1.4 *m/s*) moves the air from the centre of the validation circle outside of it. That wind speed is below the dust emission threshold. A perfect match is therefore not expected at all. Detection by both instruments means that either the emission event lasted a long time, or that it occurred in a wide enough area so that even with low altitude transport it may be observed by both instruments. This situation is more prone to occur during the evening / night comparisons because then the dust emission peak is broader in time and space,





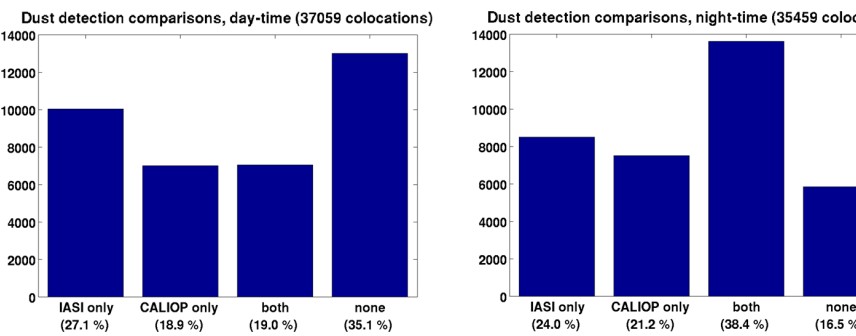

**Figure 1.** Surface dust detection within IASI-CALIOP co-locations for the whole year 2015.

mostly due to large-scale emission events. Figure 1 shows those histograms for the comparisons along the whole year 2015, for morning versus afternoon (referenced as day-time) and evening versus night (referenced as night-time). All the expected features are indeed present in those surface dust detection comparisons.

Figures 2 and 4 show the same data separated in four periods of three months matching the four seasons. Figures 3 and 5 show the geographic distribution of the co-locations and the surface dust detections. A first and direct observation is that there are more co-locations during autumn and winter than during spring and summer, and that a significant part of these co-locations occur in the western part of the Sahara (Mauritania), in the Sahel and sub-Sahel area. Only few co-locations exist in the central Sahara area, especially during the spring days. This is due to the lower coverage in that area with MAPIR, because the day-time retrievals often do not pass the post quality filters. The reason for this is still under evaluation.

During fall and winter (October to March), the co-locations show surface dust detection by both instruments mainly in the Sahel and sub-Sahel areas. In central Sahara (Algeria, Libya, north Chad), there are mostly surface dust detections with IASI in the morning (and less CALIOP detections). This is most probably due to local and short emission events occurring at the LLJs breakdown (e.g. Schepanski et al., 2009). There are significantly more surface dust detections with CALIOP than IASI in north Sudan during the night. The reason for this could be the local wind speed maximum in the middle of the night, close to the CALIOP overpass time. Those observations would therefore be local and short emission events due to the strong nocturnal winds.

During spring and summer (April to September), most day-time co-locations are located in east and north Sahara where dust emissions are rare, while night-time co-locations are more distributed. Again, significant surface dust detections occur for IASI morning measurements in central Sahara, which are not picked up by CALIOP afternoon measurements. Both these facts explain the big difference in the day-time and night-time histograms. Elsewhere, 'CALIOP only' surface dust detections occur at places close to the 'IASI only' surface dust detections (i.e. blue dots are not very far from red dots), and in-between there are places with detections by both instruments (green dots). This is probably explained by transport between overpasses, shifting the detection places. In any case, both instruments highlight mainly the same areas as dusty at the surface, with the exception of central Sahara.



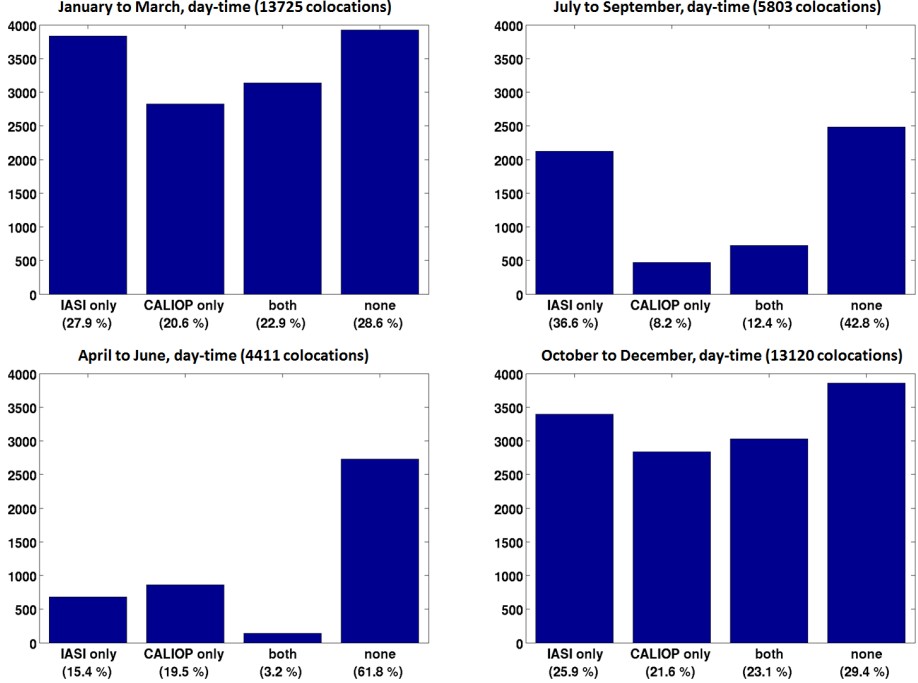

**Figure 2.** Surface dust detection within IASI-CALIOP day-time co-locations for the 4 seasons.

Although this exercise should not be considered a full validation of the MAPIR surface dust detection, it shows that those detections are reasonable. It also shows that probably some areas are under-represented in the data set. This will be taken into account in the definition of the strategy for dust emissions evaluation. In addition, this exercise allowed to determine a relevant concentration threshold for reliable MAPIR retrievals in the surface layer. This threshold of 50 *particles/cm³* will therefore be used in this work.

### 2.1.3 Use of MAPIR surface dust data

In this study, we consider monthly and yearly aggregations of data (level 3 data). To construct them, we have taken into account the specificities of the IASI and MAPIR data sets. These are (a) a reduced number of good quality retrievals over central Sahara in the summer, (b) that each cell of the level 3 grid contains a different number of level 2 pixels (mainly due to clouds), which are not equally distributed over the time period for which the data is aggregated. The latter is a more global issue that might give different weights to different days/areas and seasonal biases. To cope with these two issues, the following approach was adopted. First, we construct daily level 3 maps of quality-controlled data satisfying the surface dust detection threshold. To compute monthly or yearly aggregations, each day is given the same weight. Finally the number of days with surface dust is divided by the number of days with good retrievals to obtain a monthly or yearly fraction of days with surface dust with respect to days with good quality surface retrievals. These maps are generated with a latitude / longitude resolution of 0.5°(about 55 *km*). An example is shown in Figure 6 for the year 2015, day and night time measurements grouped, on a



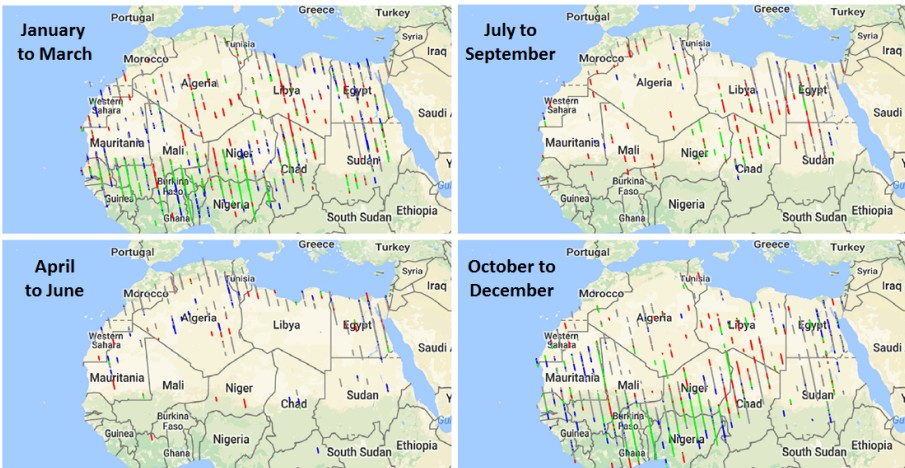

**Figure 3.** Map display of surface dust detections by IASI-CALIOP day-time co-locations for the 4 seasons. Red: IASI only; Blue: CALIOP only; Green: both; Grey: none.

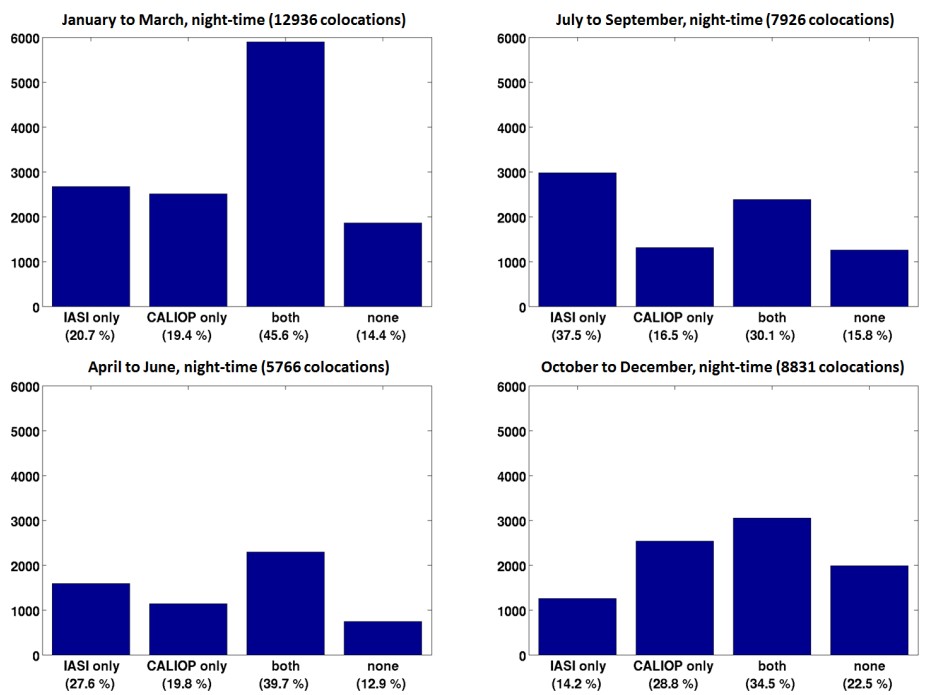

**Figure 4.** Same as figure 2 for night-time co-locations.

monthly aggregation for the north African region. A fraction of "dusty surface days" close to 1 means that for almost every day for which at least one good retrieval with surface sensitivity was available in that grid cell, there was dust detected at the surface in that grid cell. White spaces are places where no good retrievals with surface sensitivity were available. This does not



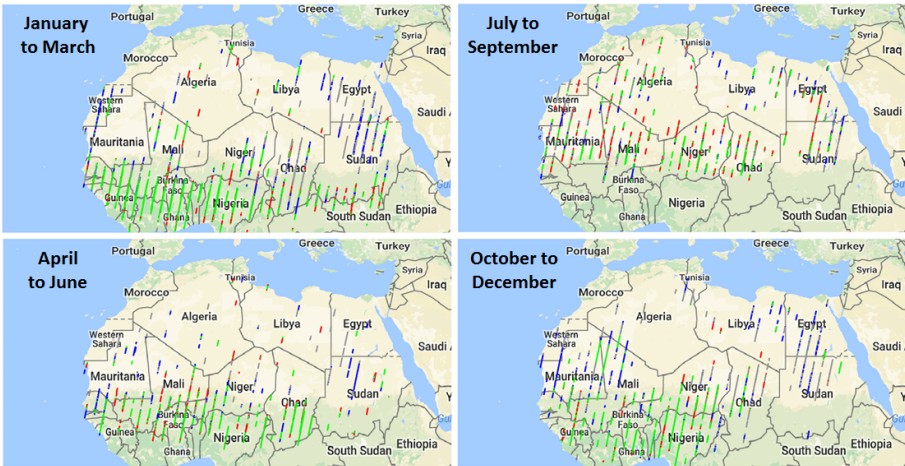

**Figure 5.** Same as figure 3 for night-time co-locations.

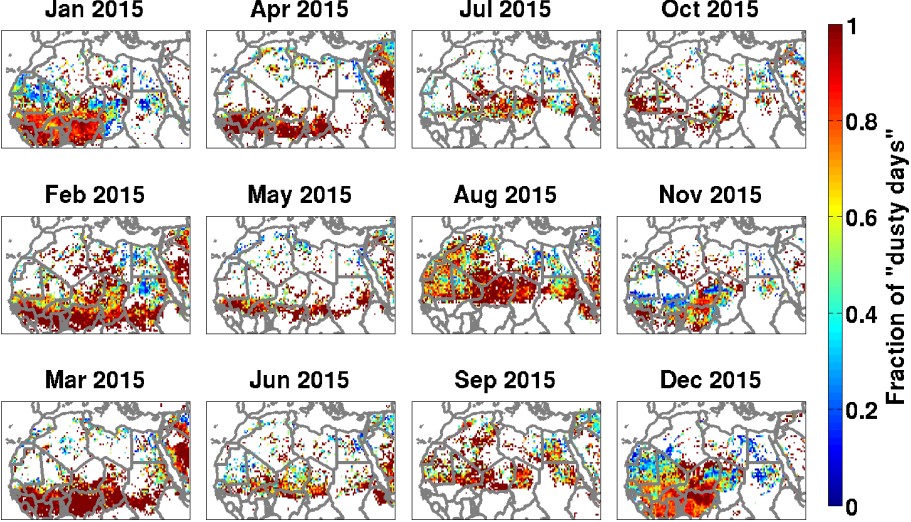

**Figure 6.** Fraction of days with surface dust from the MAPIR data set, monthly aggregation of day and night data for the year 2015.

mean that these places might not be source areas, just that the current MAPIR data set does not allow the detection of surface dust there with good confidence. However, as the analysis will show, this does not lead to missing well-known dust sources.

## 2.2 Additional data sets

The MAPIR data set allows to pin-point the dust presence at the surface. This is a significant improvement with respect to
5  dust detections in the total atmospheric column usually obtained from satellite measurements, as will be shown in section 3.1. However, dust might be present in the surface layer because it was just emitted there (or close by), but also because it is in





the process of deposition. Dust also occasionally travels at low altitudes. Therefore, we use additional data sets to evaluate if the locations where dust was found at the surface could be dust emitting. We look at the surface wind speed, land cover, soil moisture and vegetation index. For Sahel and sub-Sahel, we have also included a soil type analysis. All these parameters only help determining if the places could be sources, but none of them renders the analysis completely certain. In some specific

cases, the wind direction was also used in the interpretation.

### 2.2.1 Surface winds

In this work, we use the 10 $m$ wind velocity fields from the European Center for Medium-Range Weather Forecasts (ECMWF) ERA Interim. Our approach here is not to analyse separately each possible emission event detected using MAPIR data and confirm the presence of sufficient wind during each event. We use wind fields for a statistical analysis of the probability that

high enough winds occurred during a month. Over the Sahara this does not seem to be an issue, but over Sahel and sub-Sahel it is not straightforward because winds are less intense. In addition, Largeron et al. (2015) have studied how wind fields from global reanalyses capture the "real" observed surface wind events in the Sahelian region. They have concluded that amongst the three compared fields, ERA Interim performs best. However, all three fields show seasonal biases and systematically underestimate the strongest winds during the morning (LLJs) and during deep convective events, both being important for dust emissions.

The authors conclude that the "too low fraction of high wind speeds will lead to major errors in dust emission simulations". More specifically, using the ERA-interim 10 $m$ wind speed to model the dust emission potential can lead to underestimation of this potential by one or two orders of magnitude.

Considering this together with the emission mechanisms detailed in section 1.1, we have decided to use the wind fields as a rather low constraint for dust emission plausibility: a grid cell is considered to be a plausible dust emitter during a month if, for

at least 10% of the days, at least one of the 6-hourly ERA Interim 10 $m$ winds exceeded the threshold of 5 $m/s$. This criterion is met over the whole Saharan desert during the whole period of analysis, except for some very limited spots around the Tibesti (North Tchad, South Libya) and Tahat (South-East Algeria) mountains, during the winter. The same is observed in Morocco and North of Algeria around the Mount Atlas chain. In Sahel there is a strong seasonality of the availability of strong winds, peaking during the winter and spring, with significant yearly differences. Figure 7 shows an example of that analysis for the

year 2015. The colour scale has been selected so that grid cells in dark red are accepted.

### 2.2.2 Land cover

It is quite obvious that the land cover type is a basic criterion to discriminate plausible dust emitting surfaces. In this work, we use the land cover data from the corresponding European Space Agency Climate Change Initiative (ESA CCI) project (http://www.esa-landcover-cci.org). This data is available globally, at 300 $m$ spatial resolution, as 5-years averages for three

epochs centered on the years 2000, 2005 and 2010. As the differences in the data for the three epochs are limited to quite specific areas, and are seen only at a small scale, we could use any of the three maps for our purposes. We chose the most recent, being also the one covering years for which IASI was operating: the epoch from 2008 to 2012.





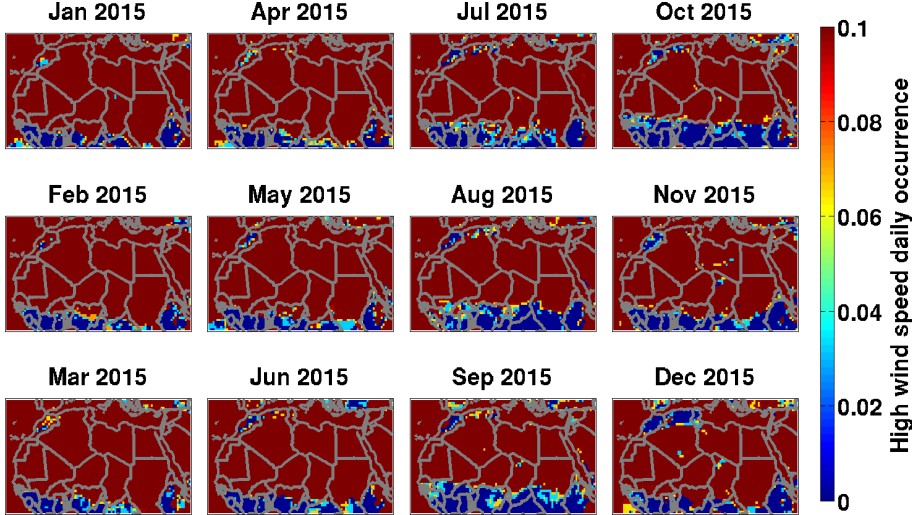

**Figure 7.** ERA-interim wind field analysis: daily occurrences of 10 *m* height wind speed higher than 5 *m/s*. The colour scale is so that the grid cells in red are accepted in our analysis.

Our selection of erodible land cover types comprises all types of bare areas, rainfed or irrigated croplands, all types of sparse vegetation or shrubland. The land cover data being a pluri-annual mean, it does not allow to catch the seasonal vegetation changes. Therefore, for all of the listed types except bare areas, additional constraints on vegetation and soil moisture are used (with their seasonality), as described in sections 2.2.3 and 2.2.4. A land cover "filter map" containing 3 values was built: not plausible dust source, plausible dust source, and plausible dust source with additional constraints (see Figure 8). The reduction of horizontal resolution from 300 *m* to our 0.5° grid was done on the basis of the filter, not on the land cover type. An area is considered to be a plausible dust source if at least 25% of its surface is of any bare area type. An area is considered to be a plausible dust source with additional constraints if at least 25% of its surface is made of any of the accepted types. In all other cases, the grid cell is considered not to be a plausible dust source.

### 2.2.3 Vegetation Index

Vegetation is known to absorb the wind momentum and prevent dust emissions. Information about the vegetation is used as additional constraint when the land cover filter requires it. That is for example the case of crops in Sahel and sub-Sahel: they are cultivated only during the wet season (June to October) and remain similar to bare areas for the rest of the year.

The vegetation data used in this work is also obtained from the ESA CCI land cover project. It is one of three global climatological weekly time series describing the natural variability of the vegetation, the snow cover and the burned areas. It is expressed in Normalized Difference Vegetation Index (NDVI). We have generated monthly data from the weekly time series, and linearly interpolated from the original grid (0.125°) to our grid (0.5°). The typical NDVI for bare areas is about 0.1. Rainfed crops in the Sahel show a NDVI of about 0.2 from December to June when no agriculture takes place, and up to 0.4 in



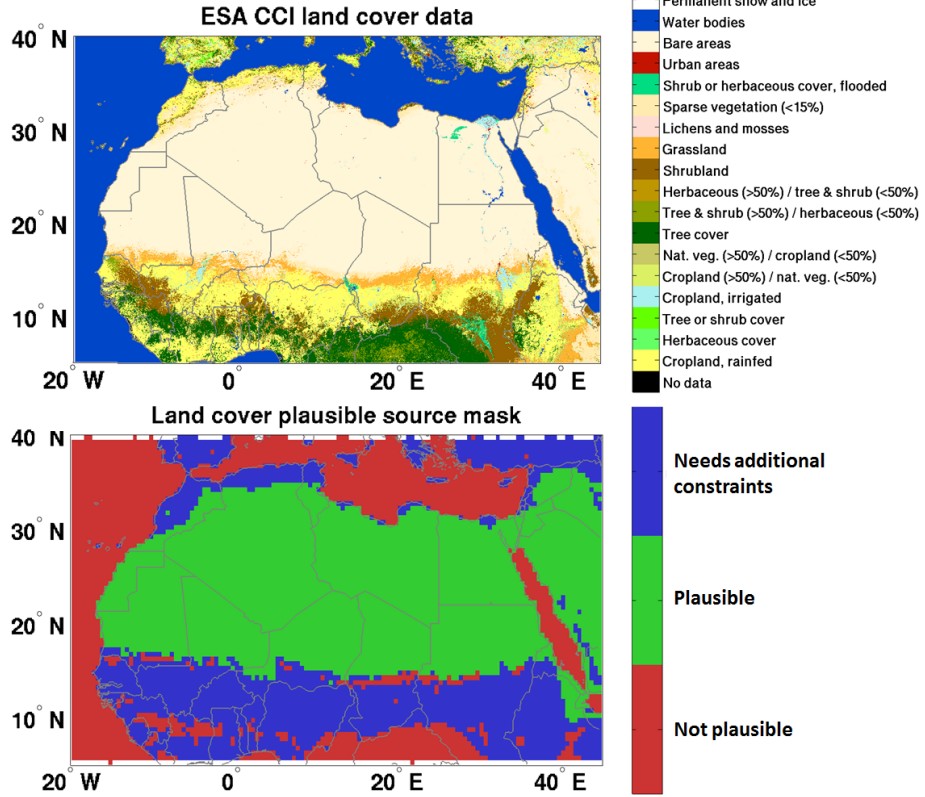

**Figure 8.** Upper plot: ESA CCI land cover data, epoch centered on 2010. Lower plot: corresponding land cover filter for plausible dust sources (see text for details).

September when the crops are fully grown. Shrublands in the South Sahel show a similar behaviour, with a NDVI close to 0.2 during the dry season, and up to 0.6 during the wet season.

Parajuli and Yang (2017) have studied the link between dust emissions and NDVI in the Bodélé depression, and concluded that dust mobilization is fully suppressed when the NDVI approaches 0.18. However, we think this was a typing error in their text, as they actually still observe locally emitted dust for NDVI up to almost 0.28 (Fig. 8e of their manuscript). Therefore we conclude that an appropriate NDVI threshold for a surface to plausibly emit dust would be 0.28. Figure 9 shows the monthly averaged NDVI from the land cover CCI, where all grid cells in dark red are rejected (NDVI>0.28).

### 2.2.4 Soil moisture

Surface soil moisture is an other important parameter to determine if dust would be able to be uplifted by winds. In this work, we use the surface soil moisture data from the corresponding ESA CCI project (http://www.esa-soilmoisture-cci.org) in its version 3.2 (Dorigo et al., 2017). That data set has been generated using active and/or passive microwave spaceborne instruments and covers the 36 year period from 1979 to 2015. It has global coverage with a spatial resolution of 0.25°, and a

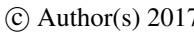



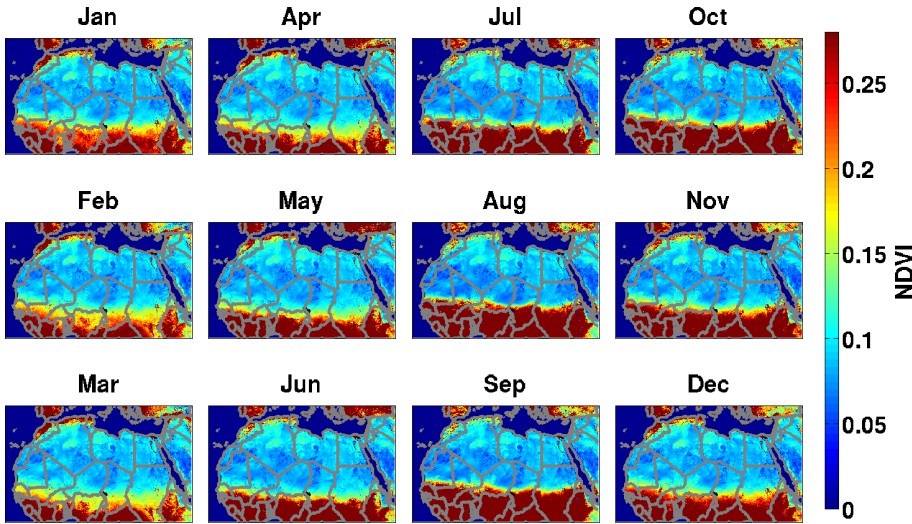

**Figure 9.** ESA CCI land cover NDVI data, monthly averages. The colour scale is so that all grid cells in red are rejected from our analysis.

temporal resolution of one day. For this work, we use the data set from the active instruments (on board ERS-1, ERS-2 and Metop-A). There are two reasons for this. First, this ensures that the soil moisture was retrieved from measurements at about the same time as the IASI overpass. Second, amongst the three data sets, the one from active measurements is the only one expressed in percentage of saturation (the other two are expressed in volumetric units $m^3 m^{-3}$), easier to use in our framework.

We have again generated monthly averages from the daily data and linearly interpolated the data from the original grid (0.25°) to our grid (0.5°). Soil moisture cannot be retrieved over tropical forests, therefore in this project we have set the soil moisture for these areas to 100%, so that they are unable to meet the criterion to be plausible dust emitting surfaces. When no soil moisture data is available for some pixels during a month, the pixel is considered to be a plausible dust emitting surface regarding the soil moisture criterion.

Kim and Choi (2015) have studied the correlation between measured dust AOD, wind speed and soil moisture. They have shown that "the threshold soil moisture for dust outbreak increased with increasing wind speed", and that for a volumetric soil moisture higher than 16% the measured AODs are only barely affected by wind speed conditions, indicating that this value could be a general soil moisture threshold above which dust emissions are almost impossible. Therefore, we consider that threshold to be relevant for our study of dust sources. Figure 10 shows an example of monthly averages for the year 2015. On

that figure, the color scale is again so that all rejected grid cells are in dark red. The seasonal cycle in the Sahel and sub-Sahel is quite clear, with the "humid limit" going North during the summer. The Sahel wet season extends mainly from June to November. There is also a clear seasonal cycle in northern Africa (Morocco, north Algeria, Tunisia, Libya).



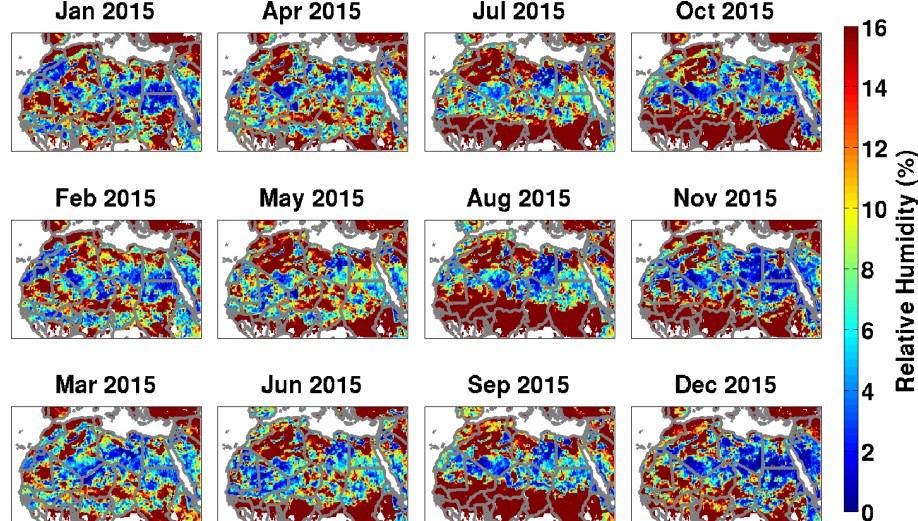

**Figure 10.** ESA CCI soil moisture data from active sensors, monthly averages for the year 2015. The colour scale is so that grid cells in red are rejected from our analysis.

### 2.2.5 Final automated filtering based on all additional data

Figure 11 shows the combination of all the previous filters for the same year 2015, showing in green the places which are plausible dust emission areas and in white those which are not plausible. There is clearly a seasonal dependence for the Sahel, which could be dust emitting only during the dry season. There is a year to year variation depending on the precise humidity and wind patterns. The seasonal variations for other regions are of smaller extent, and slightly varying with years. One of the features of these maps however looks odd: within the Sahel and only during the dry season, there is a rejected area at the borders of Niger and Nigeria, then east through Chad and Sudan. A part of this feature, in central Chad and Sudan (but not all of it) is due to the land cover mask (grassland), therefore identical for all years. The rest is due to soil moisture and varies slightly with years. These areas are not identified in our MAPIR analysis as presenting high dust loads in the surface layer during the dry season. The border between Niger and Nigeria is however separating two areas of high surface dust occurrences during the dry season.

Figure 12 shows the combination of the plausibility analysis (Figure 11) with the surface dust occurrence analysis (Figure 6). With respect to figure 6, almost all high occurrence locations south of the Sahel have been removed as considered not plausible dust emission places. However, from the comparisons with CALIOP (section 2.1.2), it is clear that the surface dust detections in that area are correct. Therefore, our conclusion is that most of the surface dust in sub-Sahel is either in a deposition process, or is transported at low altitudes. However, small-scale local dust emissions would be possible under rare occasions for which the monthly averages are not representative. To confirm this transport/deposition hypothesis, we analysed ERA-interim wind patterns. During the summer, the winds blowing towards sub-Sahel come from the ocean, explaining the absence of dust (at the surface and in the total column). During the winter, significant north-easterly winds exist (the Harmattan) with a strong surface



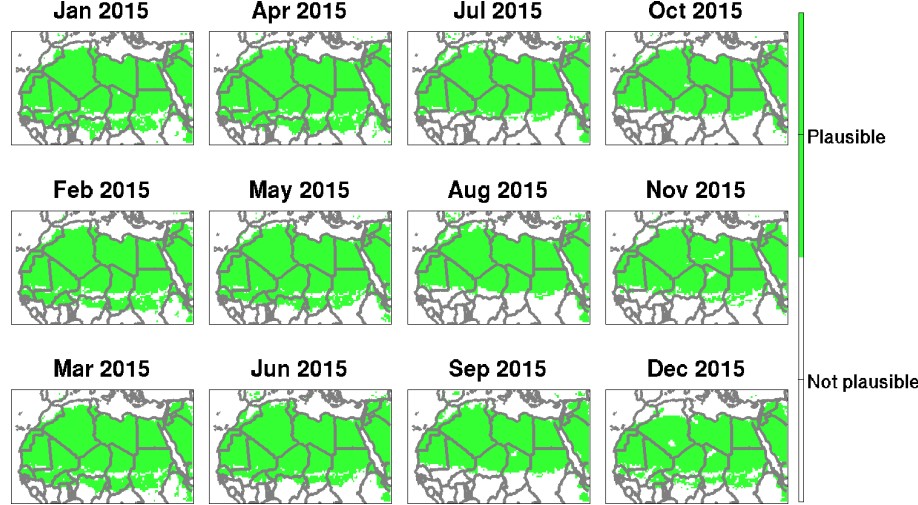

**Figure 11.** Final automated filter for the year 2015. Areas shown in green are plausible dust emitting areas, according to the series of filters discussed in the manuscript.

component that might easily transport dust at low altitudes from Bodélé. In addition, during January to March, when the largest surface dust occurrence is observed south of the Sahel, the Inter-Tropical Convergence Zone (ITCZ) actually crosses that area, trapping dust in converging winds. This is also coherent with the fact that the observed surface dust in that area seems to be spatially homogeneous, not pointing to specific emission places. Therefore, our full interpretation of the surface dust detections in sub-Sahel is that it is almost entirely due to low altitude transport from other source areas, or deposition of the dust trapped in the ITCZ. Rare local emission events could happen but would clearly not explain the observed very high occurrences of surface dust.

## 2.3 Summary of the mineral dust source analysis method

Summarizing the previous sections 2.1 and 2.2, the successive steps of the mineral dust source analysis method are:

(1) The IASI 3D dust distribution obtained with MAPIR undergoes a quality control as for any other use of the data, with an additional criterion that the retrieval must have a minimal sensitivity to the surface layer.

(2) We compute maps of dusty surface day occurrences (minimum 10 $\mu m$ AOD of 0.2 in the 1 $km$ retrieval layer above the surface elevation), aggregated monthly on a 0.5° resolution grid. Each day is given the same weight, and surface dust occurrences are provided relative to occurrences of good retrievals.

(3) We add filters pertaining to the plausibility of identified "hot-spots" to be effectively emitting dust aerosols: there must be minimum 10% of the days during which one of the 6-hourly ERA interim 10 $m$ wind speeds was at least 5 $m/s$, the land cover must permit bare areas for at least part of the year, the monthly NDVI must be lower than 0.28 and the monthly soil moisture must be lower than 16%.



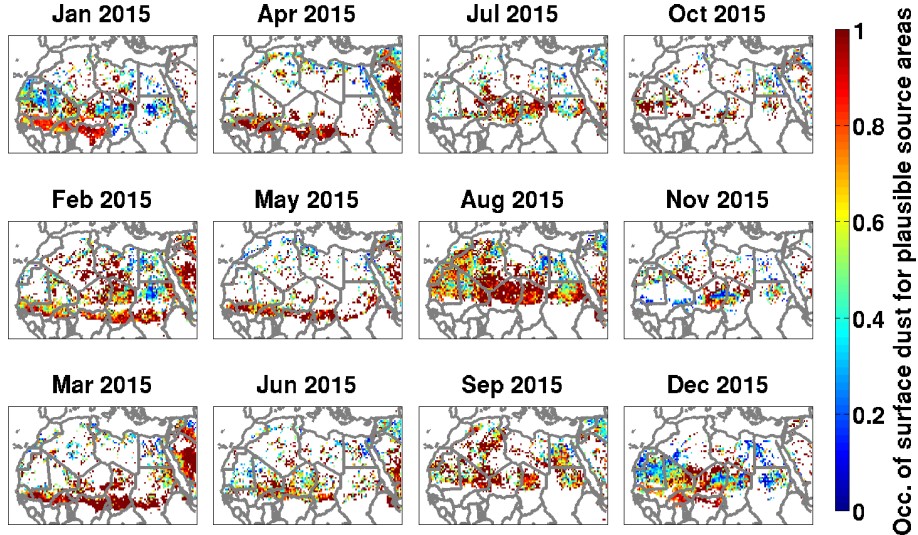

**Figure 12.** Combination of the dust emission plausibility analysis with the surface dust occurrence analysis, on a monthly basis for the year 2015.

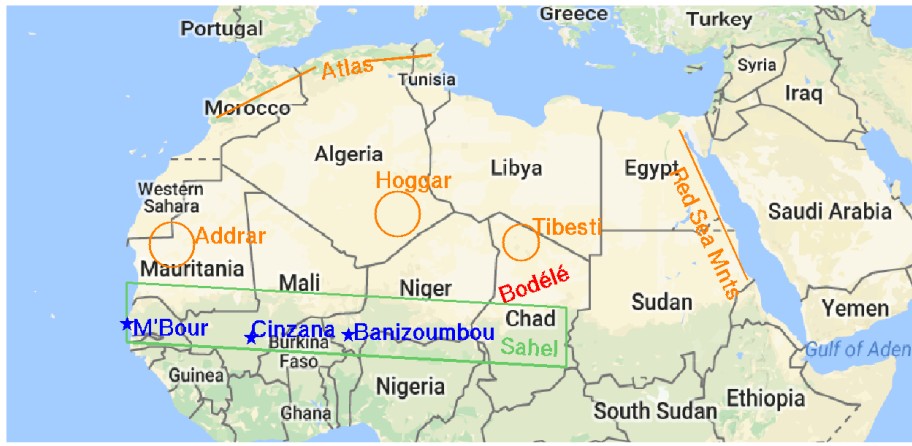

**Figure 13.** Geographical features discussed in the manuscript. Ground-based stations discussed in the manuscript are also shown.

Further in this manuscript, when nothing specific is mentioned, we refer to the plausible dust emission occurrence maps obtained with this method, such as figure 12. To help the reader best understand our analysis, we have prepared a map showing the geographical features discussed in the manuscript, and the discussed ground-based stations (Figure 13).



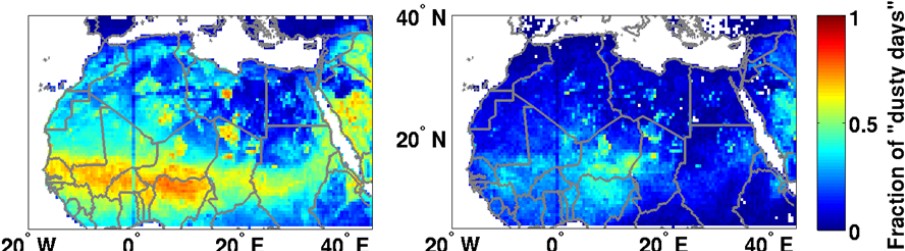

**Figure 14.** Total column (left) or surface (right) dusty days occurrences in the MAPIR data, without using the additional filtering on winds and surface properties. Data was aggregated for years 2008 to 2016.

## 3 Mineral dust source analysis: 9 years of combined data

### 3.1 Surface versus column dust detection

As mentioned in section 1.3, the dust sources studies using satellite data are all done based on total column measurements, except for CALIOP-based studies, which are then hampered by an extremely poor ground coverage. Figure 14 shows the dust occurrence in the MAPIR data set, either in the surface layer or in the total column, for an aggregation of the full data set from January 2008 to December 2016. The surface occurrences computation is done as detailed in section 2.1.3. The additional filters pertaining to surface properties and winds are not used. The column occurrence computation is done using the same method, with two differences: the quality criterion relating to surface sensitivity is removed, and the 0.2 AOD limit relates to the total column. If maintaining the quality criterion about surface sensitivity for the column dust detections (not shown), similar results are obtained but the average is noisier due to the more limited number of data satisfying conditions of good surface sensitivity. This is reassuring, because it shows that except for adding noise to the results, an analysis focusing on scenes with surface sensitivity does not bias the dust detections.

In general, the same largest yearly dust hot spots are appearing in both column and surface dust occurrence analyses but the daily occurrence of high dust load is significantly higher for column detections than for surface detections, due to the subsistence of dust in the tropospheric column after it was emitted. In addition, the whole west Sahara shows about 40% of columnar dusty days. It is likewise along the Atlas mountains, the east Mediterranean coast and over the Red Sea mountains. These areas, especially the mountains, are not known dust sources. To provide more insight into this specific point, Figure 15 shows the monthly aggregation of the full time series of column dust detections. The high occurrences of columnar dust in the mentioned areas is observed during spring and summer, with a huge peak during the summer. This corresponds to the maximum intensity of the central Saharan sources. During that period, main winds at the dust transport altitude (2.5 to 5 $km$, about 750 to 550 $hPa$) blow to the west from the central Sahara. Then, when reaching the Mauritania coast, winds continue to blow to the west leading to the transport of dust over the Ocean. If reaching the Western Sahara coast, winds are changing direction and blowing to the north-east towards the Atlas mountains, then to the east when reaching the Mediterranean Sea, in some kind of loop. Figure 16 shows the mean ERA-interim wind speed and direction for the month of June 2015 at a pressure





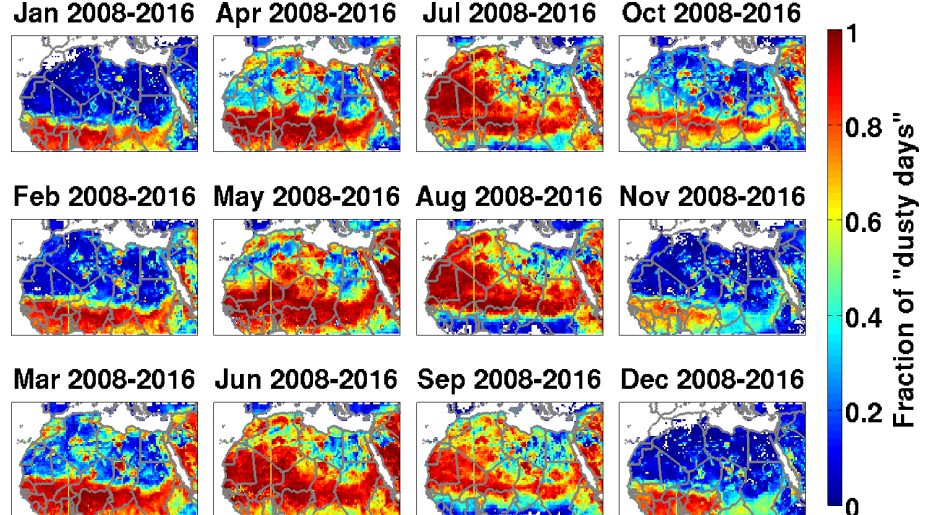

**Figure 15.** Total column dusty days occurrences in the MAPIR data, without using the additional filtering on winds and surface properties. Data was aggregated monthly for years 2008 to 2016.

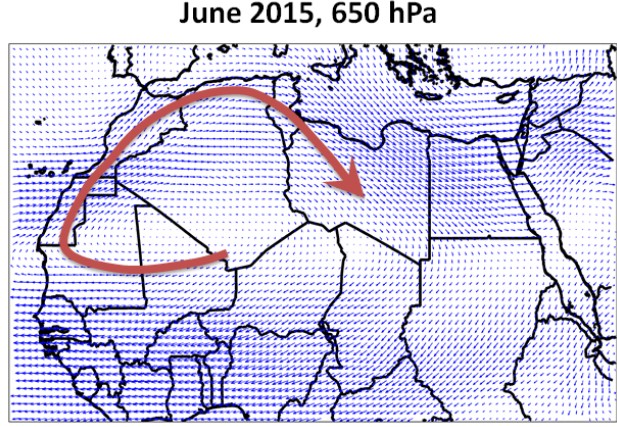

**Figure 16.** ERA-interim mean winds for June 2015 at a pressure of 650 $hPa$. The "wind loop" discussed in the text is schematically represented.

of 650 $hPa$, with a schematic representation of the loop as we described it. As the surface dust occurrence around the Atlas mountain and along the Mediterranean is low while the column occurrence is high, most of the dust in those places is almost certainly not of local origin but is transported from the central Saharan sources along a half loop. Along the Red Sea mountains, the maximum occurrence of columnar dust is recorded during spring, when the dominant winds seem to converge towards the

5    Red Sea (see figure 17). Again, most of this dust is most probably transported from Saharan sources or Bodélé.



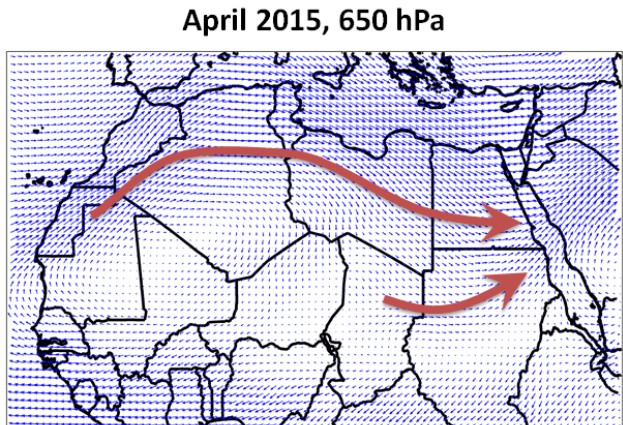

**Figure 17.** ERA-interim mean winds for April 2015 at a pressure of 650 *hPa*. The wind patterns discussed in the text are schematically represented.

From this short analysis, it is obvious that the difference between the column and surface dust occurrences relates to the distinction between dust emission and transport or accumulation. Our new method based on using a 3D dust data set is therefore fully relevant.

## 3.2 Saharan dust emission hotspots

Figure 18 shows the aggregation of the monthly full analysis for years 2008 to 2016. It clearly emphasises seasonal cycles of dust emissions, with two major "hot spot" areas, and a series of smaller ones. For the Sahara, all the additional constraints have no consequence on the analysis, as all criteria are met along the whole Sahara.

The first major surface dust hot-spot is in central Sahara, south-west of the Hoggar mountains (south Algeria, north-east Mali and north-west Niger). This area is very active during late spring and summer, especially from June to August while quiet
during the rest of the year. This is coherent with respect to recent literature information (e.g. Schepanski et al., 2007, 2012; Crouvi et al., 2012; Ashpole and Washington, 2013; Kocha et al., 2013; Evan et al., 2015; Todd and Cavazos-Guerra, 2016), even though the precise emission locations vary from one study to the other. This area is however strangely not highlighted in Gherboudj et al. (2016) as having a high dust emission potential.

The second major hot-spot is the Bodélé depression area in central Tchad, south of the Tibesti mountains, active throughout
the year with a minimum during the summer, as reported in the literature (Schepanski et al., 2007, 2009, 2012; Crouvi et al., 2012; Kocha et al., 2013; Allen et al., 2013; Gherboudj et al., 2016). Our analysis also highlights a high occurrence of surface dust west of the Bodélé depression, in east Niger. That occurrence is actually even higher than within the Bodélé depression. Gherboudj et al. (2016) show a high dust emission potential (based on simulations) covering from the Bodélé depression in Tchad to the east of Niger where we observe high surface dust occurrence. This dust emission potential also seems higher in





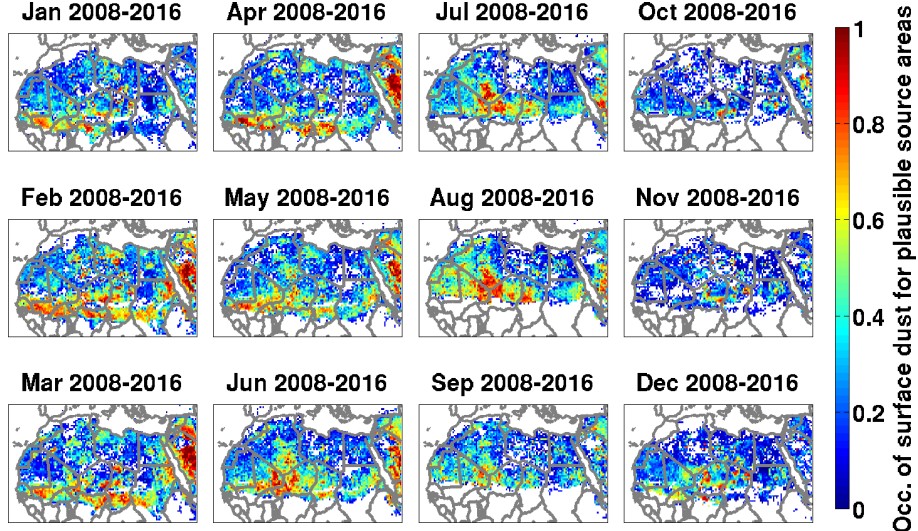

**Figure 18.** Combination of the dust emission plausibility analysis with the surface dust occurrence analysis, aggregation of monthly data for the years 2008 to 2016.

Niger than in Tchad, confirming the plausibility of significant dust emissions in that area. Significant occurrences of high dust AOD is also reported in that area, with a minimum during the summer, by Ginoux et al. (2012) and Kocha et al. (2013).

Significant occurrences of surface dust are also observed almost everywhere in the Sahara during late spring and summer, with small hot-spots in Algeria south of the Atlas mountains, in Mauritania and north-west Mali, in east Niger and in Soudan. This is again coherent with the literature (Schepanski et al., 2007, 2012; Ashpole and Washington, 2013; Todd and Cavazos-Guerra, 2016).

### 3.3 Sahel: dust emission and deposition place

Sahel is absent from most satellite-based dust sources studies. Indeed, for UV-visible instruments measuring a total AOD with no distinction on the aerosol type (as OMI and MODIS), this area is tricky as both biomass burning and dust aerosols are present. The Sahel area is however a known dust source, mostly pinpointed as due to human activities (Ginoux et al., 2012). Our analysis shows plausible dust sources in Sahel during winter and spring, the dry season. For Sahel, the additional criteria linked to winds and surface conditions take their full sense, allowing to account for the seasonal cycles of vegetation and humidity together with the significant changes in winds.

As the whole Sahel seems to be a surface dust hot spot, we have completed our analysis with soil type information, trying to pin-point locations where dust is present at the surface but could not be locally produced. Such a soil type study has been done for dust sources in Sahara by Crouvi et al. (2012), but their analysis does not cover the Sahel. Here, we used the soil types data from the European Commission African Soil Atlas (EC), based on FAO world soil maps (www.fao.org) and the harmonized soil database from the FAO and IIASA (www.iiasa.ac.at). As plausible Sahel dust emitting soils, we have selected those reported





as erodible (under dry conditions and without vegetation): acrisols, alisols, arenosols, cambisols, leptosols, lixisols, planosols and regosols.

The main Sahel surface dust hot spots in winter and spring are, from the highest to the lowest occurrences: north Nigeria (maximum in February), south-west Mali (maximum from February to May), south-western Niger (2 maxima in March and

June), south-east Tchad (maximum in March and April) and Burkina Faso (2 maxima in February and April). The north of Nigeria contains many different soil types, amongst which arenosols in the central part of the surface dust hot spot, where the occurrences are the highest. The rest of the plausible emission area contains mixed arenosols, leptosols, acrisols, lixisols and some non dust-emitting soils. South-west Mali contains again different soil types amongst which regosols and lixisols cover a significant part, matching the highest surface dust occurrence place. Some leptosols and arenosols are also reported

in south-west Mali. South-western Niger is mostly covered with arenosols. South-east Tchad is covered mainly with leptosols and lixisols. Finally Burkina Faso is mostly covered with non-erodible soil types, with small portions of lixisols and regosols. From this analysis, it would seem that the biggest surface dust host spots almost all contain significant parts of leptosols and arenosols, as do the main Saharan source areas. Regosols seem to play a role in south-west Mali. The slight hot spot in Burkina Faso reveals more difficult to attribute to local emissions. As the surface dust is observed mainly in February and April, together

with the Mali surface dust hot spot, it could easily be concluded that observations in Burkina Faso are the result of low altitude transport from neighbouring Mali. However the low altitude wind patterns in spring show a wind convergence from the north, south and east (the ITCZ) in Burkina Faso (see Figure 19), therefore excluding the hypothesis of low-altitude dust transport from Mali but allowing the hypothesis of accumulation of dust transported from sources in north-east Nigeria and in Tchad. It is however also plausible that there are small-scale local emissions in Burkina Faso during early spring, hypothesis supported

by the localised hot spots while a large area of high occurrences would be expected in case of accumulation of the transported dust. The precise surface dust occurrence hot spot in Burkina Faso is made of the vertisols type, which is a soil that reacts highly to humidity, and cracks up when dry. Crouvi et al. (2012) report their erosion potential to be higher than that of lixisols, so it would be possible that local emissions occur there under dry conditions.

This whole analysis linked to soil types does however not take into account the possibility that, as Sahel is a well-known

dust deposition place, the erosion does not occur on the local soil type but on a layer of deposited dust. The so-called local emissions from Sahel could actually be in a significant part dust from the other African sources, which has transited through Sahel with a deposition step during the summer and re-emission during the winter. If this hypothesis is correct, then any part of Sahel arid enough and without vegetation could be dust emitting, no matter the soil composition under the deposited dust layer.

As mentioned earlier, there is no previous satellite-based study of dust sources in Sahel with which we could compare our analysis. However, there is a set of three ground-based stations deployed in 2006 under the African Monsoon Multidisciplinary Analysis (AMMA). This set of stations is called the Sahelian Dust Transect. The stations are found in Banizoumbou (south-west Niger), Cinzana (south Mali) and M'Bour (west Senegal). Their location is shown in Figure 13. Using surface PM10 and wind measurements at those stations, Marticorena et al. (2010) and Kaly et al. (2015) concluded that the surface dust at

those stations during the winter is mostly due to transport from other sources, but that some local sources do exist at the end




**April 2015, 950 hPa**

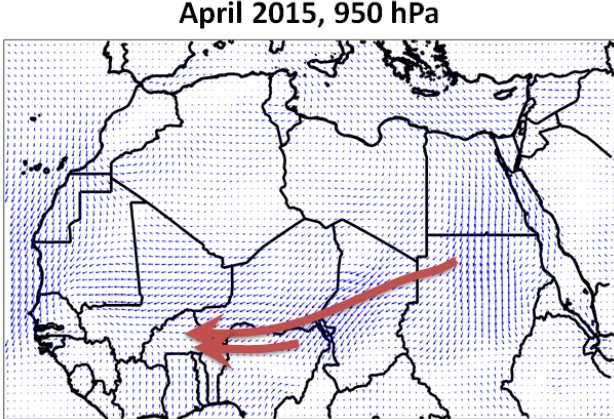

**Figure 19.** ERA-interim mean winds for April 2015 at a pressure of 950 *hPa*.

of spring (June). Those conclusions were based on the assumption of an erosion threshold of 6 *m s⁻¹*, and on the fact that the maximum of the surface dust PM10 and of the wind speed were not synchronised: surface PM10 peaks in March (secondary peak in June for Banizoumbou), while the most intense winds occur in June. However, they also mention that when looking at hourly wind data, the maximum wind speed exceeds the assumed erosion threshold of 6 *m s⁻¹* 5% of the days in M'Bour,

9% of the days in Cinzana, and 21% of the days in Banizoumbou. In M'Bour and Cinzana, more occurrences of high wind speed occur during the wet season, but the difference is only of 1 and 2% respectively. On the contrary in Banizoumbou, more occurrences of high wind speeds were observed during the dry season. In the analysis presented in this manuscript, a slightly lower wind erosion threshold of 5 *m s⁻¹* has been selected for reasons detailed in sections 1.1 and 2.2.1, and it has been assumed that if that threshold is reached at least during 10 % of the days in a month, significant dust emissions are plausible during

that month. Almost the whole Sahel meets that criterion during the entire dry season. Two of our identified Sahelian dust hot spots are around two of the Sahelian Dust Transect stations: Cinzana in south Mali and Banizoumbou in south-west Niger. Our surface dust occurrence seasonality in those places is coherent with the surface dust PM10 seasonality from Kaly et al. (2015). Therefore the discrepancy between our analysis and that of Marticorena et al. (2010) and Kaly et al. (2015) does not regard the surface dust detection but the use of winds and other ancillary data to interpret the surface dust data. The conclusions of these

two different analyses should probably be that the high surface dust occurrences in most of the Sahel are a superimposition of low-altitude transport from other Saharan sources and of local emissions, especially in the strongest hot-spots identified in this manuscript.

## 3.4   Diurnal variations

As mentioned in section 1.3, dust emissions have a known diurnal cycle and IASI measurements occur at interesting times with

respect to that cycle, close to both emission maxima. The difference between the morning and evening surface dust detections may provide information about the main activation mechanism of each emission area and/or period. Figures 20 and 21 show the





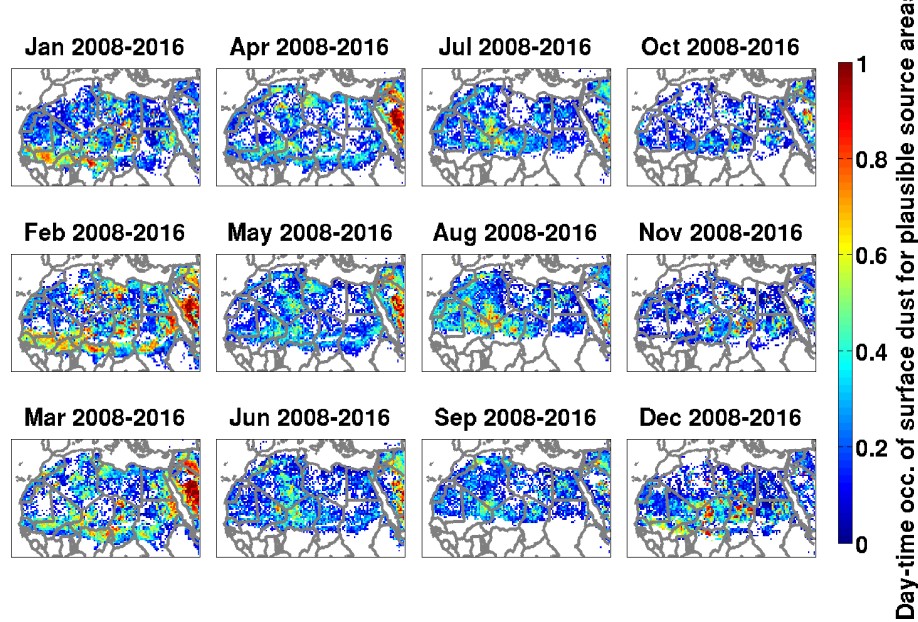

**Figure 20.** Day-time (9h30 LST) surface dust occurrence analysis, using all the additional filters as detailed in the text, for an aggregation of the data from 2008 to 2016.

surface dust occurrences for the 9 years 2008 to 2016, with all additional filters discussed previously, respectively for morning (9h30 LST) and evening (21h30 LST) IASI measurements. Our analysis will first focus on the three major hot spots identified in sections 3.2 and 3.3: central Sahara, Bodélé and the Sahel. Then other specific morning or evening emission places will be highlighted. Our findings are compared with the relevant literature.

5 In central Sahara, the pinpointed summer activity is clearly present in both morning and evening detections, with a slightly higher occurrence for evening detections, especially in the beginning of the summer (June). This is coherent with Todd and Cavazos-Guerra (2016) reporting a higher dust emission index during the night in that area, and with Kocha et al. (2013) showing dust emissions peaking in the evening in the Addrar, slightly west of the area we report here.

 The Bodélé depression seems to be more active in the morning, while the area west of it shows a high frequency of surface

10 dust during the evening. Bodélé is known to be a place with high occurrence of LLJs during the winter (e.g. Fiedler et al., 2013), coherent with those morning surface dust detections. West of Bodélé, in east Niger, the dust emission index from Todd and Cavazos-Guerra (2016) is higher during night than day, again supporting our observations. However, that dust emission potential analysis was done for summer months when emissions in that area are at a minimum (together with the occurrence of strong surface winds) and the situation might be different during the winter. From this difference in diurnal cycle for surface

15 dust occurrences in Bodélé and west of it, we conclude that there is a high probability that the area west of Bodélé is also a local source.



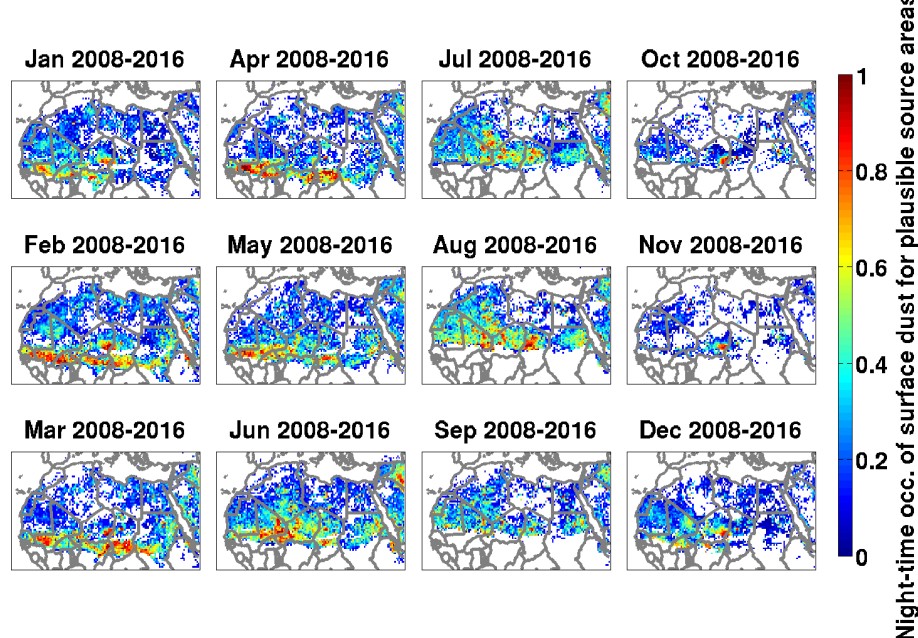

**Figure 21.** Same as figure 20 for night-time (21h30 LST).

The Sahel area does not show significant morning versus evening differences during December and January. During February to June, the whole Sahel shows higher surface dust occurrences in the evening. That difference increases from February to June, with significant surface dust occurrences in the morning in February but only very few surface dust occurrences in the morning from April on. Kaly et al. (2015) report the hourly surface PM10 along the year (averaged from 2006 to 2010) at the three

Sahelian Dust Transect stations discussed in section 3.3. In Banizoumbou and Cinzana, the peak surface PM10 occurs during morning and beginning afternoon from February to April, but significant concentrations are reported throughout the whole day. This is consistent with our detections both in the morning and evening, but not with the higher evening than morning occurrences. However, one should keep in mind that different years are averaged for both studies (2006 to 2010, or 2008 to 2016) and that a temporal evolution of the situation could occur. Then in May-June, the surface PM10 reported by Kaly

et al. (2015) in Banizoumbou and Cinzana is at a minimum in the morning and peaks mainly during the evening, being fully consistent with our observations. This whole seasonal pattern is coherent with the presence of morning LLJs during early spring, then the onset of large convective events (which mostly occur during afternoons and evenings) when approaching the wet season.

There is a significant morning hotspot in north-east Sudan, along the Red Sea, at the end of the winter (February and

March). Fiedler et al. (2013) report that area as being a LLJ hot spot from November to February, which is coherent with our observations.





There are some scattered hot spots of morning surface dust occurrences in late winter and spring in Algeria and Libya, probably due to LLJ activity.

### 3.5 Long-term evolution

When analysing the dust source occurrence maps for each year individually (figures not shown), it is obvious that there are large inter-annual variabilities. When looking at each year individually, there are less areas with information about surface dust compared to Figures 12 and 18. This is due to the limited amount of scenes with sufficient surface sensitivity for our analysis. Therefore, an analysis of inter-annual variability would require a longer time series, as will become available in the future, knowing that the IASI mission is currently planned until at least the 2020's, with a continuation later on with a new instrument with better signal-to-noise ratio (IASI-NG).

At this stage, we may only offer a basic qualitative analysis of dust emissions during the nine years 2008 to 2016. Amongst those years, 2010 is clearly the "dustier" year with the highest occurrences of surface dust. In addition, the surface dust detections in the whole central and west Sahara show high occurrences starting at the end of the winter (February) and until end of the summer (September), while usually such high generalised occurrences are detected only from June to August. Year 2009 was also a "dusty" year, with surface dust occurrences in central and west Sahara significantly larger than during the other years, but only during the summer months (on the contrary to year 2010). In Sahel, the high surface dust occurrences are observed every year, but the start and end periods of the highest occurrences vary. For example, during years 2011 and 2015, significant surface dust occurrences are detected over wide areas already in December (and they continue in January of the next year). On the contrary, during years 2010 and 2014 the Sahel surface dust occurrences are high only starting in February. The Sahel dusty season always extends until the wet season arrives (which then triggers the soil moisture filter in our analysis).

### 4 Conclusions

In this manuscript, we describe a new method for the analysis of mineral dust sources, based on the combined use of a unique dust 3D data set obtained from IASI measurements with an in-house retrieval algorithm called MAPIR, and of wind and surface state information. The method was designed to exploit best the different data sets, also accounting for their weaknesses. Monthly gridded maps of surface dust occurrences are constructed using the MAPIR 3D data set quality controlled for the specific purposes of this study. The surface dust occurrences are quantified relative to the occurrences of good-quality data. Then a complex filter based on land cover type, vegetation index, soil moisture and wind speed is used to best separate plausible emission places from places with low-altitude transported dust, or dust in a deposition process.

Our analysis clearly shows the added value of using surface dust detections instead of column dust detections. Indeed, with column dust detection, areas downwind from dust sources may easily appear as hot spots, as is the case for the whole central and west Sahara, the Atlas mountains, the Mediterranean coast and along the Red Sea mountains. The analysis based on surface dust on the other hand does not highlight those areas, and the wind patterns confirm the high plausibility of dust being transported to those areas from their sources in central Sahara rather than being locally emitted.





The analysis also shows the interest of using additional information linked to the vegetation and humidity cycles, and to the wind speed and direction. Indeed, the area south of Sahel shows very high occurrences of surface dust during the winter, but the surface and wind conditions prevent those places to be plausible dust sources on a large scale. It is possible that small-scale emissions rarely occur south of Sahel, but those would absolutely not explain the high occurrences observed. A wind

pattern analysis shows that low altitude winds converge to that area during the winter, bringing the dust from Bodélé and accumulating/depositing it south of Sahel.

The dust sources analysis over North Africa shows a good agreement with the literature, demonstrating that our method is indeed suitable for such an analysis. Our data provides for the first time a monthly aggregation over 9 years, allowing to generalise the source analysis. The latter has highlighted significant surface dust occurrences with wind and surface conditions

confirming plausibility of significant large-scale dust emissions in central Sahara, in the Bodélé depression and west of it. Small-scale dust emission places are also detected all around west and central Saraha. For all those sources, the diurnal cycle observed with our study is coherent with the literature (when it exists), and complements the available information with a more general analysis than the local ground-based diurnal cycle analyses. The source area west from the Bodélé depression is less well characterised and reported in previous studies, while from our analysis that area seems to be an important dust source.

The hypothesis of dust observed there being transported from Bodélé instead of locally produced has been analysed and our conclusion is that the probability of local emissions is high, especially when looking at the different and significant diurnal cycles in Bodélé and west of it.

The Sahel area had previously been studied mainly using ground-based data. Here, we provide a first global and detailed study of that very particular area, including an analysis of the wind patterns and soil types in addition to the dust 3D dis-

tribution, surface parameters and wind speed. This led to the conclusion that north Nigeria, south-east Tchad and south-east Mali are plausible dust emission places. On the other hand, the weak surface dust hot spot in Burkina Faso seems to be due to accumulation of transported dust. Accumulation of dust from other emission places is also possible in the whole Sahel area during the winter, due to wind convergence. Therefore, the analysis based on soil types might be irrelevant, as the deposited dust may be eroded again, no matter what soil lies beneath it. Considering, in addition to our analysis, the existing literature

on Sahel surface dust based on ground-based local measurements from the Sahelian Dust Transect stations, we conclude that most probably the Sahel dust load along the whole year is due to transport, accumulation and deposition of dust from Saharan sources, to which significant local sources add up during the dry season, especially in the hot-spots identified in our study.

A limited qualitative analysis of year-to-year variations has highlighted 2010 as being the dustier year of the last decade, with a Saharan dust season extending 2 months longer than on average, and with more intense emissions than on average. Because

of the significant year-to-year variations and the current availability of only 9 years of data, conclusions about long-term trends cannot be drawn yet with enough confidence. However, IASI and IASI-like instruments will continue to fly for a long time, and the long-term analysis will be possible in the future.

This manuscript presented a new multi-parameter method for the study of dust sources. The coherence with the existing literature gives confidence in our new method, and our analysis was therefore extended to the whole North Africa including

Sahel. This work highlights in particular the usefulness of a 3D dust distribution data set with good Earth coverage. Future



work based on this new method will be the characterisation of dust sources in the Middle-East and in Asia, which up to now have been less studied than African dust sources. In the long term, such an analysis will be possible with longer time series, allowing to study possible trends in dust emissions. The method will be easily adapted to use data from the New Generation IASI, and also most probably data from the InfraRed Sounder on-board Meteosat Third Generation (MTG-IRS).

## 5    Appendix A:  The MAPIR retrieval algorithm

The Mineral Aerosol Profiling from Thermal Infrared (MAPIR) retrieval algorithm is a significant technical and scientific improvement of the algorithm first published by Vandenbussche et al. (2013). MAPIR version 3.0 has been verified within the aerosol_cci phase 2 project, in a IASI dust AOD round robin experiment (Popp et al., 2016). After that experiment, the algorithm was improved to correct some of the issues discovered during the round robin experiment, and then used in its
version 3.5 to analyse the full IASI/Metop-A data set, from 2007 to 2016. That data set is used in this manuscript, therefore MAPIR version 3.5 is described here. An adapted version of MAPIR has been used to study the 3D distribution of volcanic ash aerosols after the Puyehue Cordón Caulle eruption in June 2011 (Maes et al., 2016). MAPIR v3.5 is currently under validation within the aerosol_cci project.

### A1    Radiative transfer considerations

The MAPIR algorithm operates on TIR radiances, in the spectral range called the atmospheric window because of its high transmittance (in absence of clouds and aerosols), between 800 and 1300 $cm^{-1}$ (12.5 to 7.7 $\mu m$). At those wavelengths, mineral aerosols show distinctive extinction signatures (e.g., Di Biagio et al., 2014; Klüser et al., 2012; Vandenbussche et al., 2013). The main one is due to the Si-O stretching and has the form of a V-shape centered between 1000 and 1100 $cm^{-1}$ (10 and 9 $\mu m$). The exact wavelength of the peak extinction depends on the mineralogical composition of the aerosols. Additional distinctive
(while weaker) extinction signatures are observed in the TIR spectral range, which also depend on the exact mineralogical composition of the aerosol. In measured TIR spectra, those signatures may appear in emission (so-called inverse V-shape) in particular cases when the thermal contrast between the aerosol layer and the surface is highly positive (the aerosols being warmer than the surface). In that case, the aerosols thermal emission may compensate for both absorption and scattering.

The MAPIR algorithm relies on Lidort (Spurr, 2008), a very accurate radiative transfer code including multiple scattering.
Radiative transfer calculations are performed inline for each retrieval. The forward modelling is computed with a 0.25 $cm^{-1}$ spectral step and with a Gaussian instrument line shape (0.5 $cm^{-1}$ full width at half maximum), such as to properly reproduce the sampling and resolution of level 1c IASI data. Three spectral windows are used for the retrievals, avoiding the huge ozone absorption band centered at 1040 $cm^{-1}$ (9.6 μm): 905 to 927 $cm^{-1}$, 1098 to 1123 $cm^{-1}$ and 1202 to 1204 $cm^{-1}$. The noise level is set to 4 $10^{-7}$ $W\ cm^{-2}\ Sr^{-1}\ cm^{-1}$, which is about seven times higher than the instrument's spectral noise in that window. This value
was empirically selected as the lowest value allowing retrievals to converge for a selection of dusty scenes with good ancillary data. It is necessarily higher than the instrument noise in order to account for additional "noise" due to model uncertainties (all non-retrieved parameters described hereunder), which are currently neither quantified nor taken into account otherwise. Lidort





computations are set-up with 4 quadrature streams in the cosine half-space, using the delta-m approximation for the aerosols phase function, and under the assumption of lambertian surface. Refraction is not included because of the nadir geometry. Solar sources are not relevant in the selected spectral windows.

The Lidort radiative transfer code is linked to a Mie code preparing the aerosol optical properties from their particle size distribution and refractive index. Mineral dust aerosols are thus parametrized as spherical particles in MAPIR. The real shape of those aerosols highly departs from a sphere (e.g., Laskina et al., 2012), but this has only minor implications in the TIR spectral range for coarse mode particles, compared to the impact of the Particle Size Distribution (PSD) and Refractive Index (RI) for which significant uncertainties exist. In MAPIR, we use the Gestion et Etudes des Informations Spectroscopiques Atmosphériques (GEISA, Jacquinet-Husson et al., 2011) and HIgh-resolution TRANsmission database (HITRAN, Massie, 1994; Massie and Goldman, 2003) dust-like RI based on measurements by Volz (1972, 1973) and Shettle and Fenn (1979). MAPIR uses a log-normal PSD with median radius of 0.6 $\mu m$, geometric standard deviation of 2, corresponding to an effective size of 2 $\mu m$ (e.g., Clarisse et al., 2010).

**A2  Retrieval scheme and parameters**

The MAPIR retrieval scheme is the Optimal Estimation Method (OEM, Rodgers, 2000), which iteratively adjusts a state vector, composed of 7 variables: the surface temperature (Ts) and the vertical profile of dust aerosols concentration, from 1 to 6 $km$ altitude in steps of 1 $km$. The spectral derivatives with respect to the state vector variables are computed during the first and last iterations. One of the strengths of the OEM is that is also produces averaging kernels, which enable to analyse the sensitivity of the retrievals at the different altitudes, and the correlation between the retrieved parameters. The OEM is an iterative algorithm requiring an a priori value of the state vector. For Ts, we use different data sets depending on the date. For retrievals on IASI data after 14 September 2010, we use Ts from the European Organisation for the Exploitation of Meteorological Satellites (EUMETSAT) IASI operational Level 2 (L2) retrieval, version 5 and beyond (the version depends on the date, no full reprocessing is currently available). For data prior to 14 September 2010, the Ts retrieved by the EUMETSAT IASI operational L2 algorithm version 4 or prior has shown significant issues (many unrealistic values), in particular for desert surfaces. Therefore, for dust retrievals at those dates we use the Ts from the European Center for Medium-Range Weather Forecasts (ECMWF) ERA Interim 6-hourly reanalysis as a priori. The Ts a priori standard deviation is set to 5% in all retrievals. This might seem high but the diurnal Ts variation reaches more than 20K for desert surfaces. In addition, the Ts retrieved within the IASI operational processing is obtained without considering the presence of aerosols, and is therefore expected to be significantly biased in case of high mineral aerosol load.

As a priori for the vertical profile of desert dust concentration, we use the LIdar climatology of Vertical Aerosol Structure (LIVAS) monthly 1°x1°climatology, obtained from CALIOP data (Amiridis et al., 2015). That climatology contains mean vertical profiles of dust extinction at 1064 and 532 nm, with high vertical resolution. These extinctions were converted to vertical profiles of dust particles number concentration at the vertical resolution of the retrieval, using the visible (532 $nm$) cross-section of the aerosol particles used in MAPIR (using the mean radius as single particle size). To account for the fact that CALIOP measurements are sparse, and therefore the continuity of the climatology amongst adjacent 1°x1°cells is not ensured



(the mean extinction in adjacent cells may come from measurements on different days), we compute the dust a priori profile as a horizontal running mean over 25 cells (5 in latitude, 5 in longitude). If, after that, there remain cells for which no LIVAS data is available or the extinction is reported to be 0 (which is not an acceptable value for the a priori), we use the mean extinction over the whole geographic area (0 to 40°N; 80°W to 120°E). This situation is extremely rare. The standard deviation for the

dust aerosol vertical profile is set to 100% at all altitudes and all locations to reproduce the high variability of the dust aerosol load. The a priori vertical profile is assumed to have a Gaussian correlation length of 1 $km$.

The dust AOD is obtained by integration of the dust profile to a total column, and conversion using the extinction cross-section at the desired wavelength.

### A3  Ancillary parameters

To complete the surface parameterisation for TIR emission, the algorithm requires the surface emissivity in addition to Ts. For ocean, the surface emissivity is close to 1, with only a slight spectral and spatial variation. We use the published sea surface emissivity of Newman et al. (2005), as constant over time and space. For land surfaces, the emissivity has a significant spectral dependence on surface type and varies slowly as a function of time, depending on the surface composition, humidity, vegeta-tion,... We decided to use the monthly emissivity climatology from Zhou et al. (2011), obtained from IASI measurements. For

places where dust is present during a long period, e.g., close to major dust sources, this emissivity might be biased because no clear-sky observations were available.

The mean surface altitude of each scene is extracted from the National Oceanic and Atmospheric Administration (NOAA), Marine Geology and Geophysics topographic data.

To finalise the parameterisation required for the radiative transfer, it is also necessary to describe the atmospheric state. In

that regard, the two most important parameters for our retrievals are the vertical profiles of atmospheric temperature and water vapour. Those are taken from IASI L2 operational products from EUMETSAT; the processing version number depends on the date of the data set (quality described in August et al. (2012)). Other relevant atmospheric gas profiles ($CO_2$, $O_3$, $N_2O$, $CH_4$ and $HNO_3$), for which the accuracy is less important, are taken from the US AirForce Geophysics Laboratory (AFGL) tropical climatology (Anderson et al., 1986). All gases line parameters come from the HITRAN 2012 database (Rothman et al., 2013).

Continua are computed using the MT_CKD 2.5 formalism (Mlawer and Tobin, 2012).

### A4  Data preparation

Prior to undertaking the MAPIR retrievals, the IASI data are filtered. A first basic filter removes spectra containing negative radiance values in the spectral range needed by MAPIR. Then only scenes with less than 10% cloud fraction from the EU-METSAT IASI operational L2 cloud product are retained for the dust retrievals. Unfortunately, that product seems to misflag

some intense dust clouds as meteorological clouds, removing that data from our analysis. The design of a dedicated cloud flag within MAPIR would require non-negligible additional developments and is currently not considered.

When the surface altitude in a scene is higher than one (or more) retrieval altitude(s), the dust concentration at that (those) altitude(s) is set to 0 prior to retrieval (and cannot deviate from this value during the retrieval).




## A5  Quality filtering

A first general quality control on the retrieval results tests the goodness of the fit: the root mean square of the spectral residuals (between the measured spectrum and the modelled spectrum after the retrieval) must be lower than 2 $K$ over land, 1 $K$ over sea. In addition, the total AOD at 10 $\mu m$ should be at most 8. This latter test might reject some very rare extremely intense

dust events, but allows to reject retrievals under cloudy conditions (which were mis-flagged in the EUMETSAT IASI cloud product and therefore used in MAPIR for dust retrievals). In those cases, MAPIR either fails or retrieves an extremely dense (unrealistic) dust cloud that would have a similar radiative impact as a meteorological cloud in the spectral windows used in MAPIR.

*Author contributions.*  S. Vandenbussche developed the MAPIR algorithm, performed the analysis presented in this manuscript and was the

main writer. M. De Mazière supervised the work and reviewed the manuscript.

*Competing interests.*  There are no competing interests pertaining to this work.

*Acknowledgements.*  This work could not have been done without the necessary data sets. We therefore greatly acknowledge the EUMETSAT and EUMETCast service for IASI data used in the MAPIR retrievals, the NASA Atmospheric Science Data Center and the CNES for CALIOP data used in part of the validation, the ESA CCI land cover and soil moisture projects, and ECMWF for providing the ERA-interim wind

fields used in this study. The MAPIR developments were supported by the Belgian Science Policy supplementary researcher program, and by the European Space Agency (ESA) as part of the Aerosol_cci project. The specific dust source study was carried out under a programme of, and funded by, the European Space Agency. More specifically under a Living Planet Fellowship project entitled "A new method for assessing mineral dust sources using vertical profile information retrieved from IASI radiances" (MIDUSO). We also acknowledge A.C. Vandaele, N. Kumps, E. De Wachter, V. Letocart, O. Rasson and the BIRA-IASB ICT team for their involvement in the MAPIR scientific developments,

the IASI data analysis and the establishment of the MAPIR processing chain. Sophie Vandenbussche personally acknowledges S. Plummer for his relevant suggestions to successfully lead this analysis.




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
