# Peer review of "African mineral dust sources: a combined analysis based on 3D dust aerosols distributions, winds and surface parameters"

_Atmospheric Chemistry and Physics, 2017_

## Referee Comment (RC1) · Anonymous Referee #2 · 30 Nov 2017

The authors present the distribution of North African dust source regions from a combination of IASI measurements with information on winds, land cover, vegetation and soil moisture. The results are of interest to the scientific community but are presented in a poor way and with little reference to the wider field. The manuscript requires major revisions before it can be considered for publication in ACP.

**Major issues**

- It is not common to have sub-sections in the Introduction. I would also suggest

to shorten the Introduction to the relevant information. The details presented in Section 1.3 could be moved to Section 2 where intsruments and methods are being described. Also, please provide the details on MAPIR in the main text rather than the Appendix. I found it confusing to read about results without any description of the used algorithm.

- I suggest to find a better name for what you call "surface layer" right now to avoid confusion with the actual surface layer. The lowermost kilometer or lowermost layer would do.

- From reading Section 2.1.1 it is not clear what quality control measures are being used. Overall, this section is not very focused and could be shortened.

- Section 2.1.2 also is not very focused and does not provide crucial information: Which region is being compared? What is meant with best comparison statistics? What is meant with dust detection for CALIOP and IASI. Is it only based on AOT or would CALIOP, for instance, need to show dust or polluted dust in the feature mask? The information in Figures 1, 2 and 4 would be much better presented in a Table that gives the percentage of the respective detection. Altogether, it is not clear what value is provided by the comparison between MAPIR and CALIOP surface dust detection if it is not quantitative. It is my impression that the information provided in this section could easily be conveyed in a paragraph. Figures 3, 5 and 6 should be omitted or moved to the supplementary material.

- Section 2.2.1: Do you check in any way that the days of increased wind speed are the same as the days with MAPIR dust detection in the lowermost layer? Could there be a bias due to a mismatch of high-wind-speed days and dust days? Figure 7 seems unnecessary as your criterion is fulfilled throughout most of the study region.

- Figures 7 to 12 should be omitted or moved to the Appendix or the Supplement

as they are hardly being referred to in the text, and thus, seem unnecessary. It would be better to go straight to the findings for the full time series rather than focusing on a single year.

- At some point in the paper, it would be good to provide an overview map of known dust sources (could be combined with the information in Fig. 13) and a discussion on how dust sources are being defined in dust transport models. This would then allow to compare these source locations to your findings and point our possible impacts of your work.

- I don't see the point of Section 3.1. The discussion doesn't seem to be within the scope of the paper. Also, not a single reference is being provided regarding circulation pattern over north Africa. I suggest to omit this section and the related Figures 15 to 17.

- Section 3.2 provides an overview of the dust sources. I am missing discussion of what makes those regions good dust sources. Also, a schematic map of the commonly known sources and new sources identified in this study would be good.

**Minor issues**

- please avoid colloquial language such as good agreement or good representation.

- page 2, line 1: I have heard of a semi-direct effect but not of semi-indirect ones.

- page 2, line 3: Do you mean the dust particles act as cloud condensation nuclei (CCN) and affect cloud droplets or as ice nucleating particles (INP) and affect cloud ice?

- page 2, lines 7-9: This paragraph could be omitted.

- page 3, line 4: Replace first six words in line with Below

- page 4, line 13: change exploited to operated?

- page 14, lines 3-7: Please contact the authors to clarify instead of writing a speculative paragraph.

- page 22, line 18: What are FAO and IIASA?

- page 28, line 18: What is meant with a global study of that area?

- page 28, line 23/24: You are contradicting your own work and it would be worthwhile to investigate the effect of not accounting for soil type in your analysis.

- Author contributions: From the description it seems to me that S. Vandenbussche should be the sole author of this paper. Supervision and reading a manuscript don't warrant co-authorship.

---

## Referee Comment (RC2) · Anonymous Referee #1 · 20 Feb 2018

The paper by Vandenbussche and De Maziere presents an analysis of desert dust sources over North Africa from satellite data and some ancillary filters. The particularity of this study is that is uses satellite retrievals of vertical profiles of dust derived from IASI thermal infrared spectra. Therefore, satellite data of dust load near the surface is estimated which is an advantage with respect to previous satellite-based studies of North African dust source using column-integrated amounts or lidar transects with very coarse daily coverage. Therefore, the approach is interesting and potentially provides new information. However, several major revisions are needed in order that the paper

is publishable in the ACP scientific journal. These major revisions are first listed, then important modifications that are also needed and other minor points that should be modified.

I strongly recommend the following major revisions, so that the paper is publishable:

1) The title: The paper only analysis dust over North Africa and not the whole continent. It does not really use 3D dust distributions but surface retrievals (the authors mention this explicitly in page 5, lines 30-31). The use of winds and surface parameters is very limited in the paper to be explicitly mentioned in the title (see below for further comments on this). I strongly recommend to change the title of the paper as: "North African mineral dust sources: a combined analysis based on surface dust detections and ancillary data" or similar.

2) Quality of the figures: readability of most monthly figures is poor, with very small panels, noisy data and very difficult to distinguish the evolution for a relatively small region (e.g. Bodélé depression). Their quality should be revised and improved.

3) MAPIR data coverage: It is clear from Figure 6, that MAPIR retrievals are not available for each cloud free scene. For a given month, MAPIR data is rather limited, covering in most cases half of North Africa and rarely the northernmost part of Africa. This is of course not only linked to cloud cover, since IASI data is mostly available twice a day and clouds over the Sahara do not persist along a whole month. Moreover, MAPIR retrievals detect dust is in most cases (at least two thirds) and in a rather limited number of cases the retrieval detects dust-free scenes. So, to which extend, dust detection is linked to MAPIR data coverage? Therefore, it is important to clearly show and quantify the coverage of MAPIR data as a fraction of all possible measurements. Caution should be clearly point out for regions with limited data coverage (e.g. Northernmost part of Africa?).

4) Atmospheric dynamics over North Africa: The analysis of transport patterns in the current paper is too simplistic. One cannot simply draw out transport patterns around

a large continent (over distances greater than 5000 km) by using a map of average monthly winds at a pressure level. No reference to previous work is done. The main dynamical actors of the region, largely know in literature, are not mentioned (African Easterly Jet, African Easterly Waves, Sub-tropical westerly Jet, Inter-tropical front, etc). This can only be addressed by dedicated studies using trajectories or transport/dispersion models and considering the 3D aspects of transport. Unless fully revised and properly addressed, I strongly recommend withdrawing Figures 16, 17 and 19 from the paper and the corresponding comments.

5) Surface wind and moisture filters: Monthly estimates of surface wind speeds and soil moisture are not directly linked to dust uplift, but their instantaneous values (in hourly scales). Surface winds can be very strong a few times a month (for example due to Mesoscale Convective Systems) and uplift large quantities of dust. However, this region may not pass the filter of a frequency higher than 10% of 5 m/s. Soil moisture is highly variable in time. It can evaporate very fast (in a few hours) in the first centimetres of soil during daytime over the desert and emit large quantities of dust. Only after a rain event, it clearly inhibits dust uplift but only a few hours later, it dries out and dust can be very easily removed. Only daily or sub-daily estimates of surface winds and soil moisture are useful for determining uplift potential. I really recommend revising the criteria used for this filters and use daily estimates of surface wind and soil moisture in coincidence with the actual satellite data, otherwise these filters do not have much physical sense.

Important revisions:

6) A description of other retrievals of dust using IASI measurements should be provided in the introduction of the paper.

7) Agreement or not between MAPIR and CALIOP detections of dust at the surface: Figures 3 and 5 show very large discrepancies between IASI and CALIOP. Mainly one region of coincidence is observed: The Sahel. These differences cannot only be

explained by the time of the day of the measurements since once dust uplift occurs in a given region, most dust remains suspended nearby for more than 5 hours. Please, clarify the discrepancies in a more thoroughly analysis.

8) Dust in deposition process (e.g. line 15, page 16): Dry deposition of Aeolian dust always occurs when close to the surface. There is no sense in spotting a particular place as a region for "dust in deposition process". One can tell that the region is not a source region, but dust is transported across.

9) Accumulation of dust after transport: the concept of accumulation of dust after transport is strange. An atmospheric constituent may accumulate at a given region if there is no wind after emission. However, a dust plumes is uplifted by winds and then it is transported and dispersed in the atmosphere. Air masses do not stop at a certain region after transport, but they are diluted horizontally and vertically by mixing and dust burden can reduce also due to continuous dry deposition and wet deposition when raining. Please clear out this aspect or use another term.

10) The northern part of Sahel is a place pointed out as a dust source by Middle and Goodie (2001) and Israelevich et al. (2002). Differences of current results with respect to their works should be clearly given.

11) Since it is a key and uncertain parameter: A sensitivity test of the MAPIR approach with respect to surface emissivity as a function of the location over North Africa should be given. Imprecise emissivity may cause geographical biases in the region selected as dust sources.

12) Figure 10 show numerous regions with soil moisture above 16% but not suppressed from the "all filters" picture in Figure 11. Please clarify.

13) Figure 13 only gives very approximate positions of mountains and regions. It is not a proper style for a scientific publication. It should be revised.

14) Please verify English language and many typing mistakes all across the

manuscript.

Minor revisions:

15) Page 5, line 3: "each cloud-free IASI spectrum". According to Figure 6, this is not the case since many regions are not covered by MAPIR data, which of course, are not covered by cloud all month long.

16) Land cover data (section 2.3.2 and Figure 8): it only points out desert areas. Very little selection is done. Please, clarify this.

17) Page 14, line 5: Citing a reference for NDVI and telling there is "a typing error in their text" seems strange to point out. A different reference should be used.

18) Writing style: Some expressions seem as oral language: "it is reassuring", " a huge peak".. "some kind of loop" "area is tricky". These terms should be revised.

19) Page 1, line 20: dust is located below 7 km because of their size only? It is because of their sources and mechanisms to mix it in the atmosphere... Ash is directly ejected at elevated layers and they are even coarser in size.

20) Page 1, line 24: absorption features? Better to use "absorption bands".

21) Page 3, lines 1-3: not clear, which mechanism accounts for 1%? What happens with the 99% remaining?

22) Page 3, lines 9-11: Notion of dry and moist convective events should be given, as well as other mechanisms (extra-tropical cyclones/cold fronts, meteorological cold fronts).

23) Section 1.2: Page 4, line 9-10: Only the studies are mentioned and not the results. This should be more precise.

24) Page 6. Lines 20-21: statement "this is true only if those 2 parameters are independent..." is not clear. This should be better explained
25) Page 7, lines 23: the threshold for CALIOP AOD is the same for daytime and nighttime? Signal-to-noise ratios are very different in these two cases. For certain, CALIOP cannot measure AOD as low as 0.05 during the day. This threshold seems very low even during the day. What is the accuracy for CALIOP derived AOD using in other studies like Todd and Cavazos-Guerra (2016)?

26) Figures 1, 2 and 4: histograms are not very informative.

27) Page 17, line 4: This is unclear "dust trapped in the ITCZ" what does it mean? What is this mechanism?

28) Page 21, lines 14-20: The Bodélé depression hotspot is not clearly seen in Figure 18, nor the region east of Niger? The quality and size of the images do not allow to easily recognizing this spot.

29) Page 22, lines 14-20 & Page 23: lines 1 - 15: this soil data analysis is not clear. Conclusions are difficult to understand. The analysis should be supported by a figure showing the regions with the different soil types and confronted with dust maps. Why such dataset is not used as filter in section 2?

30) Page 23, line 16: The ITCZ is not expected to be convergence from the north, south and east at a given country. It is a large-scale structure, which changes in position every day, and it is closely linked to the Inter-Tropical front.

31) Page 23: lines 24-29: Transport and deposition of Saharan dust over the Sahel, which is afterward uplifted in a different season: This hypothesis is based on which scientific evidence? Are these speculations? If so, why they are mentioned?

32) Page 24: lines 13-16: "The conclusions of these two different analysis should probably be…" Here the authors of the manuscript justify a discrepancy with the conclusions from two other papers by telling that their dataset should have been interpreted in a different way. This is strange and awkward for a scientific paper. One can tell conclusions from a published dataset, but cannot change the conclusions from other

scientists.

33) Page 25: line 9: "The Bodélé depression seems to be more active in the morning " where specifically is this shown in Figures 20 and 21? The Bodele region (with marked limits) should be much better identify in the figures.

34) Maps are very noisy and it is difficult to know where red spots are in the same particular region from one to the other.

35) Page 25 line 14: "the situation might be different during winter" why this statement is not clear? The datasets shown in the paper do not show this?

36) Page 25: lines 14-16: Why conclusions on a source region west of Bodélé are linked to those from Bodélé itself? This is statement is not clear and should be better explained.

37) Page 26: line 4: The sentence is not clear. Re-write it please.

38) Page 26, lines 11-12: Consistency with LLJs during early spring is not clear. This should be explained in a much clear way.

39) Section 3.5: The title of this section is not clear. It should be named "Inter-annual evolution" or similar. This section draws conclusions from a dataset that is not shown. Evidence for these statements is not given. Therefore, either this section should be withdrawn or clear figures showing this inter-annual evolution should be presented.

40) Page 27, line 21: "unique" means that only the MAPIR approach derives dust 3D data? This is not the case. Please correct.

41) Page 27, line 27: "dust in deposition" is always occurring when close to the surface. There is not privileged place for this. Therefore, this cannot be specified as such.

42) Page 27, line 30-32: The analysis of monthly average winds is not sufficient for this statement of transport from central Sahara.

[Figure]

43) Page 28, lines 1-4: Large dust emissions over the Sahel occur often by very strong winds (sporadic and possible missed by the filter) associated with Mesoscale Convective Systems.

44) Page 28, line 16: "probability of local emission is high" how is this probability measured? It is quantified?

45) Page 28, line 18: "global". This analysis is not global (worldwide). The term is not correct.

46) Page 28, line 35: "good Earth coverage" this is not fully correct since it is not the case for single years as the example of 2015.
* * *

---

## Author Comment (AC1) · 19 Feb 2019

We would like to acknowledge the referees for their work and their very relevant suggestions for improvement of the manuscript. We would also like to apologize (to the referees and the scientific community as a whole) for the long delay before answering the referees' comments and proposing a revised manuscript. During the first reviewing period, we have discovered an unsolvable issue in the MAPIR version 3.5 used in the first manuscript, and started the development of a new version. That new version 4.1, based on a different radiative transfer code, has quickly shown a great potential for

improvement of the MAPIR data and of this study.

To ensure the best possible quality of the analysis and with agreement of the editors, we have decided to use that new version for our revised manuscript. There was therefore a substantial delay due to the processing time for the full IASI data time series with this new algorithm. A publication was just submitted (Callewaert et al, AMTD 2019) describing that new algorithm in detail, with detailed validation, helping make this manuscript shorter and more "to-the-point" of the sources study. The main differences in MAPIR v4.1 (revision) versus v3.5 (first manuscript) are described hereunder before the answer to the referees comments. In the manuscript, the full algorithm description (Appendix) is now removed and replaced by a much shorter algorithm description in the main text (section 2) with reference to the full paper. As most of the developments of the new MAPIR version were done by our colleague Sieglinde Callewaert, we would like to add her as co-author of the revised manuscript.

As a consequence of obtaining this new data set, and of a different improved use of the wind, soil moisture and vegetation data as suggested by Referee 1, most of the results analysis had to be redone and obviously rewritten. Large changes are therefore done to the manuscript.

**MAPIR v4.1 versus MAPIR v3.5**

The table hereunder summarizes the main differences between the algorithm versions. The modification that drove all changes was the change of radiative transfer model after the discovery of a problem in the Jacobian calculation for low emissivity surfaces in lidort v2.7. We could have updated to the most recent version of lidort, which does not have that Jacobian issue anymore, but in our specific application that newest version of lidort was much slower than the previous version, making it impossible to handle in terms of computing time and cost. Therefore we moved to using RTTOV, and it

revealed to be very efficient. As interfacing with RTTOV is much different and much easier than interfacing with lidort, we also wrote a separate retrieval code dedicated to our applications instead of using the more general retrieval code ASIMUT as before. This allowed us to adapt more easily the retrieval itself, therefore we added the Levenberg Marquardt regularization, and we now perform the retrieval in the logarithm of the concentration (instead of concentration), avoiding problems of mathematically plausible negative concentration values, which are not accepted by the radiative transfer codes. We also had to modify the state vector to layer concentrations instead of levels, as this is how RTTOV works (in ASIMUT, the retrieval was made on levels, but the radiative transfer in Lidort is also done with layer properties; now these conversions are removed as we do the retrievals directly on layer concentrations). The increased sensitivity with the new method allowed to lower the lowermost retrieval altitude. The validation of the new MAPIR v4.1 data shows no more AOD overestimation (which was a known issue of MAPIR v3.5), but still a dependence in the quality of the input temperature profiles (we use those from EUMETSAT IASI level 2 product). That dependence is expected as the temperature profile is a crucial parameter for the aerosol profiles retrieval in the thermal infrared. It leads to more noisy aerosols data, with the presence of some outliers, for periods with the lowest quality of the EUMETSAT IASI level 2 data (especially its version 4, for the period before 14 September 2010). All inputs to the retrievals are maintained identical to the previous MAPIR version: IASI level 1 spectra, level 2 profiles of temperature and humidity, all aerosol parameters and ancillary data.

For the dust source study in this manuscript, using the new MAPIR v4.1 data provides better data in general, but especially better surface sensitivity and therefore more coverage for the lowermost layer dust. See also the answer to the Major revision 3 from Referee 1.

| MAPIR v3.5 | | MAPIR v4.1 |
|---|---|---|
| Lidort v2.7 | **rad. trans.** | RTTOV v12 |
| Optimal Estimation | **retrieval** | Optimal Estimation + Levenberg Marquardt |
| aerosol concentration and Ts 6 levels 1:1:6km | **state vector** | log(aer. conc.) and Ts 7 layers centered 0.5:1:6.5km |
| AOD overestimation, "noisy" bad Jacobians if low surface emissivity | **known issues** | Dependence with T and H2O profiles quality |

**Answers to referee 1**

Major revisions

1) The title: The paper only analysis dust over North Africa and not the whole continent. It does not really use 3D dust distributions but surface retrievals (the authors mention this explicitly in page 5, lines 30-31). The use of winds and surface parameters is very limited in the paper to be explicitly mentioned in the title (see below for further comments on this). I strongly recommend to change the title of the paper as: "North African mineral dust sources: a combined analysis based on surface dust detections and ancillary data" or similar.

We do agree that the analysis does not make use of the full 3D dust distributions. However, the retrieval algorithm delivers vertical profiles, from which we select only the lowermost layer for this analysis. If the title mentions only "surface dust detections", it could mislead the reader to thinking that we used in situ surface measurements or any method sensitive only to the surface dust. We have improved strongly the use of winds and soil moisture as suggested by the referee (see answer to comment 3). We propose

to adapt the title as following: "North African mineral dust sources: a combined analysis based on 3D dust aerosols distributions, surface winds and ancillary parameters"

2) Quality of the figures: readability of most monthly figures is poor, with very small panels, noisy data and very difficult to distinguish the evolution for a relatively small region (e.g. Bodélé depression). Their quality should be revised and improved.

The noisy appearance of some monthly figures was not linked to the way those are plotted but to the data itself. This is improved in the new manuscript because there is a better coverage with the new MAPIR version 4.1 (see answer to the next comment). The monthly figures are indeed a bit too small but this was the maximal size of the ACP template. After verification with the editor, we have increased the figure size to a full page width for those monthly figures. In addition, we provide, in Appendix B, 5 figures that zoom on the areas pinpointed as dust emission hot-spots in the results analysis.

3) MAPIR data coverage: It is clear from Figure 6, that MAPIR retrievals are not available for each cloud free scene. For a given month, MAPIR data is rather limited, covering in most cases half of North Africa and rarely the northernmost part of Africa. This is of course not only linked to cloud cover, since IASI data is mostly available twice a day and clouds over the Sahara do not persist along a whole month. Moreover, MAPIR retrievals detect dust is in most cases (at least two thirds) and in a rather limited number of cases the retrieval detects dust-free scenes. So, to which extend, dust detection is linked to MAPIR data coverage? Therefore, it is important to clearly show and quantify the coverage of MAPIR data as a fraction of all possible measurements. Caution should be clearly point out for regions with limited data coverage (e.g. Northernmost part of Africa?).

The MAPIR retrievals are run for each cloud-free scene. There were some failing and lots of bad quality retrievals over deserts in version 3.5 used for the first manuscript. But what explains the low coverage of the surface detection is the requirement that the MAPIR retrieval has sensitivity to that surface layer (through the averaging kernel analysis explained section 2.1.1 of the first manuscript, or section 2.1.4 of the revised manuscript). That was mentioned in the manuscript (bottom of page 10 in the first manuscript: "White spaces are places where no good retrievals with surface sensitivity were available"). If one looks at the total column plot (Figure 15 in the first manuscript), for which the criterion on surface sensitivity was not applied, the coverage is very close to being full. As the referee points out, mostly dusty scenes met the minimal surface sensitivity criterion, which was the reason for a qualitative analysis only. In the new MAPIR version 4.1, the surface sensitivity is highly enhanced therefore the surface dust analysis coverage is much better. Some areas still remain without surface sensitivity, but in those areas, at the periods where there is no surface sensitivity, no dust emission is expected. The updated manuscript insists more on the missing data areas due to the lack of surface sensitivity, and a plot of the monthly number of good retrievals was added (figure 1 in the revised manuscript) in the section describing the quality control

linked to surface sensitivity (section 2.1.4 of the revised manuscript).

4) Atmospheric dynamics over North Africa: The analysis of transport patterns in the current paper is too simplistic. One cannot simply draw out transport patterns around a large continent (over distances greater than 5000 km) by using a map of average monthly winds at a pressure level. No reference to previous work is done. The main dynamical actors of the region, largely know in literature, are not mentioned (African Easterly Jet, African Easterly Waves, Sub-tropical westerly Jet, Inter-tropical front, etc). This can only be addressed by dedicated studies using trajectories or transport/ dispersion models and considering the 3D aspects of transport. Unless fully revised and properly addressed, I strongly recommend withdrawing Figures 16, 17 and 19 from the paper and the corresponding comments.

The concept of average wind direction has already been used to interpret dust transport in general over North Africa (see for example Figure 2 in Schepanski 2012 doi:10.1016/j.rse.2012.03.019 or Figure 1 in Todd 2015 doi:10.1016/j.atmosenv.2015.12.037). However we do agree with the referee that our attempt to analyze transport patterns was very limited. Both referees had a similar remark with respect to that, therefore we followed the suggestion to remove that part of the analysis.

5) Surface wind and moisture filters: Monthly estimates of surface wind speeds and soil moisture are not directly linked to dust uplift, but their instantaneous values (in hourly scales). Surface winds can be very strong a few times a month (for example due to Mesoscale Convective Systems) and uplift large quantities of dust. However, this region may not pass the filter of a frequency higher than 10% of 5 m/s. Soil moisture is highly variable in time. It can evaporate very fast (in a few hours) in the first centimeters of soil during daytime over the desert and emit large quantities of dust. Only after a rain event, it clearly inhibits dust uplift but only a few hours later, it dries out and dust can be very easily removed. Only daily or sub-daily estimates of surface winds and soil moisture are useful for determining uplift potential. I really recommend revising the criteria used for this filters and use daily estimates of surface wind and soil moisture in coincidence with the actual satellite data, otherwise these filters do not have much physical sense.

We do agree that the best analysis would be with wind and soil moisture data at times of IASI observations. Our initial choice to use monthly averages was linked to the availability of the surface wind data (ERA-interim) every 6 hours. It was impossible to use surface winds at the time of the IASI overpass without unreasonable assumptions about the wind evolution between model time stamps. In addition, as discussed in the manuscript (section 2.2.1) the ERA-interim 10-m wind speeds have been shown to be underestimated in the Sahara and Sahel. Using a statistical approach (with a low threshold) seemed, under those conditions, a good way to determine if an area was potentially a dust emitting area. The same approach was therefore used for the soil moisture analysis to give some coherence to the full analysis. The intention was not, due to those limitations, to specify each surface dust detection separately as plausible emission event, but in general to specify an area as plausibly emitting area, and obtain only qualitative information. That was clearly stated a number of times in the manuscript.

However, due to the delay needed for updating our MAPIR data, it happens that a new version of the wind model data is also available (ERA-5), with hourly data. We have thus updated our analysis with the use of those hourly 10m-winds, the new criterion being that the wind at the model time stamp before or after the IASI overpass time must exceed the threshold of 5m/s for the observed surface dust to be considered locally emitted. Of course, we have also updated our use of the soil moisture data, now using daily estimates (and the updated version of the data). For the soil moisture data, as already mentioned in the manuscript, we have selected the data obtained from satellites with overpass time similar to that of IASI or even on board the same satellite platform as IASI. However, the data is provided as daily average, with no ability to separate the morning and evening overpasses. We have also updated the use of the NDVI data to use the best temporal resolution provided, i.e. a weekly climatology (it was previously also averaged over a month for consistency).

The change in the use of surface winds has reduced everywhere the fraction of lower-most layer dusty days. The filtered occurrences are in general at least 2 times smaller than the unfiltered ones, and some areas have almost no more filtered occurrences with the new wind filter (especially in Sahel). For the evening IASI overpass, it would seem that the model winds are underestimated. This is discussed in section 3.3 in the new manuscript.

The soil moisture filter in itself (after applying the winds filter) has much more limited impacts, but the new way to account for soil moisture does modify the filtered surface occurrences in some areas of the Sahel, as does the NDVI filter.

Qualitatively, the same source areas are however still highlighted in the Sahara. In Sahel, the areas highlighted as plausible local sources are significantly reduced. This shows indeed that soil moisture and wind criteria have to be used very carefully in that sensitive area, and that the previous version of the manuscript highlighted too many "plausible source areas". Quantitatively, for example the Bodélé depression is now much more clearly seen as a prominent source during the winter, and the Sahel source

areas are less dominant. The whole section analysing the sources, and especially the Sahel part, has therefore been modified.

Important revisions

6) A description of other retrievals of dust using IASI measurements should be provided in the introduction of the paper.

We do not think that in a publication exploiting the information retrieved by one IASI algorithm we should describe all other IASI algorithms. The main purpose of this manuscript is not the algorithm description but the use of the data, and no other IASI algorithm is currently capable to deliver similar data as what is used here, i.e. vertical profiles for the complete IASI time series. The introduction is already quite long, as indeed pointed out by the second referee, and we believe that maintaining the description of source mechanisms and climate effects of dust is more in the scope of this manuscript than describing the other existing algorithms which do not provide useful data for this study. We have included a short sentence and references at the end of the section on the added value of using IASI (section 2.1.3 in the new manuscript): "The other dust aerosols retrievals from IASI either provide global coverage and full time series but no vertical profiles (refs) or do provide vertical profiles but no full time series nor global coverage (ref)".

7) Agreement or not between MAPIR and CALIOP detections of dust at the surface: Figures 3 and 5 show very large discrepancies between IASI and CALIOP. Mainly one region of coincidence is observed: The Sahel. These differences cannot only be explained by the time of the day of the measurements since once dust uplift occurs in a given region, most dust remains suspended nearby for more than 5 hours. Please, clarify the discrepancies in a more thoroughly analysis.

It is not so certain that dust would remain at least 5 hours in the source area in the 1km-layer above the surface (which is what we compared). Indeed, dust uplift occurs with a surface wind speed threshold of 5m/s (18km/h), meaning that dust can travel about 100km in 5 hours if such a wind remains. The distance criterion for a co-location between IASI and CALIOP was set to 50km in order to try to detect the emission events at a similar horizontal resolution as the one used in the sources study (0.5°). Increasing the co-location distance to 100km would result in even bigger uncertainties as to the relevance of the comparisons (are the same events observed by both satellites?).

With the MAPIR version 3.5 used for the first manuscript, indeed most of the co-locations with CALIOP were in the Sahel area because of the lower coverage of data with surface sensitivity in Sahara.

In the revised manuscript, considering that now there is another manuscript describing the algorithm with detailed validation (Callewaert et al, AMTD 2019), we decided to remove those attempts at comparing surface dust detections. This is also because close to sources, the time difference of 3 to 5 hours between IASI and CALIOP is not acceptable. Indeed, the atmospheric conditions could be completely different, with for example a dust emission event occurring at the IASI overpass, and completely finished at the CALIOP overpass time, leaving only the elevated dust layer. Comparisons are therefore always subject to assumptions, and bad comparisons are extremely difficult to interpret: are they due to bad results, or was it a short event not seen by both instruments? In the algorithm description and validation manuscript, we used groundbased lidar data, and the CATS instrument on board the International Space Station. Those allow for better time co-locations (we selected maximum 1 hour), and therefore comparisons without the need of assumptions about possible plume transport. Those comparisons show a very good capability of MAPIR to detect dust plumes and reproduce their vertical extent, included in the lowermost layer.

8) Dust in deposition process (e.g. line 15, page 16): Dry deposition of Aeolian dust always occurs when close to the surface. There is no sense in spotting a particular place as a region for "dust in deposition process". One can tell that the region is not a source region, but dust is transported across.

We agree. See also next answer. This paragraph was removed due the reorganization of the manuscript, but the remark was kept in mind for other places with similar discussions.

9) Accumulation of dust after transport: the concept of accumulation of dust after transport is strange. An atmospheric constituent may accumulate at a given region if there is no wind after emission. However, a dust plumes is uplifted by winds and then it is transported and dispersed in the atmosphere. Air masses do not stop at a certain region after transport, but they are diluted horizontally and vertically by mixing and dust burden can reduce also due to continuous dry deposition and wet deposition when raining. Please clear out this aspect or use another term.

Accumulation is used to refer to the specific situation where the dust is transported away from sources, then remains in suspension at a place of wind convergence, where therefore the dust concentration may stay high or even increase with time. This is for example highlighted by Klose et al (GRL 2010) for Sahel, where they also use the words "dust accumulates over the Sahel". Reference to that publication was added in

the discussion of Sahel results in the new section 3.2 (as the paragraph from previous page 16 was removed due to manuscript reorganization).

10) The northern part of Sahel is a place pointed out as a dust source by Middle and Goodie (2001) and Israelevich et al. (2002). Differences of current results with respect to their works should be clearly given.

A reference to the work of Middleton and Goudie 2001 (doi:10.1111/1475-5661.00013) was added in the introduction of section 3.2 discussing Sahel and previous results based on satellite data. Israelevich et al. 2002 (doi:10.1029/2001JD002011) studied North African dust sources with no analysis of Sahel; a reference to their work was added in section 3 where listing previous work.

11) Since it is a key and uncertain parameter: A sensitivity test of the MAPIR approach with respect to surface emissivity as a function of the location over North Africa should be given. Imprecise emissivity may cause geographical biases in the region selected as dust sources.

Indeed, imprecise surface emissivity could cause biases in the retrieved vertical profiles. However, such a sensitivity test would require a separate full study of its own (and therefore a separate manuscript). Indeed, the effect of an uncertainty in the surface emissivity on the retrieved dust aerosol profile depends on the atmospheric conditions (temperature profile), the surface temperature and the dust aerosol load itself. We have added a sentence to the manuscript mentioning this possible bias (section 2.1.2, with the short algorithm description: "This climatology was obtained from IASI measurements. For places where dust is present during a long period, e.g., close to major dust sources, this emissivity might be biased low because no clear-sky observations were available.").

12) Figure 10 show numerous regions with soil moisture above 16% but not suppressed from the "all filters" picture in Figure 11. Please clarify.

Those are areas classified as arid in the land cover filter, therefore the soil moisture additional filter was not used there (green area in Figure 8 in the first manuscript version). (The additional vegetation filter was also not used there). In those areas, even if the humidity was high in the monthly average, they could dry very fast and still reasonably be dust emitting places if wind conditions allow it. In the revised manuscript, we maintained this separation of bare areas for which the criteria on humidity and vegetation are not used, and the other plausibly emitting surface types, for which the criteria on humidity and vegetation are used. One reason is that most retrievals used to obtain the humidity and vegetation information work less good over bright areas (at visible wavelengths) and their results are therefore more uncertain.

13) Figure 13 only gives very approximate positions of mountains and regions. It is not a proper style for a scientific publication. It should be revised.

We have modified that figure, now based on a google map with annotations (new Figure 5).

14) Please verify English language and many typing mistakes all across the manuscript.

Done. The manuscript had been passed through a dictionary before submission but we did an additional check.

Minor revisions

15) Page 5, line 3: "each cloud-free IASI spectrum". According to Figure 6, this is not the case since many regions are not covered by MAPIR data, which of course, are not covered by cloud all month long.

Same answer as to comment 3.

16) Land cover data (section 2.3.2 and Figure 8): it only points out desert areas. Very little selection is done. Please, clarify this.

Indeed, the selection is meant to be quite conservative, so all bare areas are considered to be OK (no additional filter needed on humidity or vegetation), and all partially vegetated areas (partially in space or time) are considered to be potentially OK, if they pass the vegetation and soil moisture filters.

The full list of accepted types was not inserted in the manuscript as this is really technical.

Bare areas (green in Fig 8 of the first manuscript, Fig. 4 of the revised manuscript): classes 200 (Bare), 201 (Consolidated Bare) and 202 (Unconsolidated Bare)

Areas for which we used the additional filter (blue in Fig 8 of the first manuscript, Fig. 4 of the revised manuscript): classes 10 (Rainfed cropland), 20 (Cropland irrigated or post-flooding), 120 (Shrubland), 121 (Shrubland evergreen), 122 (Shrubland deciduous), 130 (Grassland), 150 (Sparse vegetation (tree shrub herbaceous cover) (<15%)), 152 (Sparse shrub (<15%)), 153 (Sparse herbaceous cover (<15%))

This was summarized in the previous manuscript in the first lines of page 13 by "[...] erodible land cover types comprises all types of bare areas, rain-fed or irrigated croplands, all types of sparse vegetation or shrubland". We have now added grassland (which fills the strange red line in Sahel in the previous Figure 8) and added this to the summarized description, together with the fact that sparse vegetation means less than 15%. This occurs in the revised manuscript, section 2.2.2.

17) Page 14, line 5: Citing a reference for NDVI and telling there is "a typing error in their text" seems strange to point out. A different reference should be used.

During personal discussion with the first author (Sagar Parajuli), clarification was provided about that NDVI analysis and the content of the figure with respect to the text. The data between 0.18 and 0.28 NDVI is very noisy and does not show a decreasing trend. Therefore, we agree to our mistake and we corrected to a threshold of 0.18 in NDVI. The effect of this change is limited to small areas in the Sahel, and is not seen in the comparison of the monthly maps but only in a difference map. So the change does absolutely not modify the general conclusions drawn from the analysis, although it does change slightly the numeric values of the surface dust detections after filtering.

18) Writing style: Some expressions seem as oral language: "it is reassuring", " a huge peak".. "some kind of loop" "area is tricky". These terms should be revised.

Agreed and done.

19) Page 1, line 20: dust is located below 7 km because of their size only? It is because of their sources and mechanisms to mix it in the atmosphere: : : Ash is directly ejected at elevated layers and they are even coarser in size.

Indeed. We have modified by "due to their emission mechanism and size".

20) Page 1, line 24: absorption features? Better to use "absorption bands".

Done.

21) Page 3, lines 1-3: not clear, which mechanism accounts for 1%? What happens with the 99% remaining?

Suspension, the direct uplift of fine particles by strong lift, is responsible for only 1% of the total uplift. The rest is due mainly to the saltation mechanism (mentioned a few lines before), and a little bit to creeping mechanism (mentioned also a few lines before). The sentence has been clarified to "The third emission mechanism, suspension, occurs when fine particles are directly uplifted by strong winds and is reported to account for only about 1% of the total dust emissions (Gherboudj et al., 2016).".

22) Page 3, lines 9-11: Notion of dry and moist convective events should be given, as well as other mechanisms (extra-tropical cyclones/cold fronts, meteorological cold fronts).

Those 3 lines were the introduction to the paragraph, which then contains more details on the low-level jets (lines 3 to 16) and the convective events (lines 16 to 26). Cyclones, haboobs (humid), dust devils (dry), are mentioned as examples of those convective events. We do not intend to go more in details on those mechanisms here, it is out of the scope of this manuscript, but the main types of convective events are listed.

23) Section 1.2: Page 4, line 9-10: Only the studies are mentioned and not the results. This should be more precise.

This manuscript is not intended as a review of all previous methods and results. In addition, we can't reproduce the figures from previous publications so we can't really mention the results except by making a list of source areas, which is not as precise as a map and not friendly to read. In addition, each study was done using a different instrument, and different time periods, leading to slightly different results. The introduction to the section 3 was rewritten to be clearer in that sense.

24) Page 6. Lines 20-21: statement "this is true only if those 2 parameters are independent: : :" is not clear. This should be better explained

The number of degrees of freedom (DOF) in a retrieval is the number of independent pieces of information originating from the measurements (as defined line 13 of page 6, first manuscript). If there are 2 DOFs, it is common to assume that the 2 pieces of information which can be retrieved are the total column (or AOD) and the mean altitude. However, in TIR retrievals, those are not independent and therefore do not represent the 2 independent pieces of information. Two partial columns or concentration at two altitude levels would represent those 2 independent pieces of information, if selecting them properly (accounting for the values in the diagonal of the averaging kernel). As this is too technical and not useful to the content of this manuscript, we have decided to removed that sentence, not because it is not true but because it would indeed require more detailed technical explanations to be fully understood by the reader.

25) Page 7, lines 23: the threshold for CALIOP AOD is the same for daytime and nighttime? Signal-to-noise ratios are very different in these two cases. For certain, CALIOP cannot measure AOD as low as 0.05 during the day. This threshold seems very low even during the day. What is the accuracy for CALIOP derived AOD using in other studies like Todd and Cavazos-Guerra (2016)?

This section is removed from the revised manuscript. To quickly answer the comment: In Winker et al, 2013 (doi: 10.5194/acp-13-3345-2013), it is stated that "CALIOP appears to be adequately detecting aerosol for mean extinctions greater than 0.005 to 0.01 km–1". In a 1km-layer (which is what we had done), that would lead to an AOD as low as 0.005 to 0.01. In Omar et al, 2013 (doi: 10.1002/jgrd.50330), it is concluded that CALIOP underestimates AOD when AOD is on the order of 0.05 or lower (AERONET comparisons on continental sites). This was attributed primarily to a failure to detect weak aerosol layers. So we do believe that our selected threshold was realistic.

26) Figures 1, 2 and 4: histograms are not very informative.

Those are not anymore part of the manuscript, as the CALIOP comparisons section was removed.

27) Page 17, line 4: This is unclear "dust trapped in the ITCZ" what does it mean? What is this mechanism?

It means dust remaining in suspension in the area of the ITCZ, which is over Sahel during end winter and early spring. It has been highlighted in Klose et al, 2010 (doi: 10.1029/2010GL042816) that the Sahel Dust Zone is a "convergence zone of dust transported from the Saharan interior" and 3 different mechanisms have been highlighting explaining how dust is transported to Sahel and why is remains there, in some

cases for several days before it is transported westwards or deposited. We have added that reference and relevant information from it in the manuscript.

28) Page 21, lines 14-20: The Bodélé depression hot-spot is not clearly seen in Figure 18, nor the region east of Niger? The quality and size of the images do not allow to easily recognizing this spot.

The new figures are larger and less noisy due to the better quality of the new MAPIR version. We hope that this allows a better reading of those figures. Inserting a mark for the Bodélé depression (or other features) in the plots themselves would mask the events and is therefore not really an option. We now provide in Appendix B zooms on the dust emission hot-spots discussed in the text.

29) Page 22, lines 14-20 & Page 23: lines 1 - 15: this soil data analysis is not clear. Conclusions are difficult to understand. The analysis should be supported by a figure showing the regions with the different soil types and confronted with dust maps. Why such data set is not used as filter in section 2?

The soil analysis was removed from the revised manuscript, following comments from both referees. Indeed, as we had discussed in the first version of the manuscript, dust is deposited in the Sahel during the winter, when it is transported from Sahara and Bodélé, therefore any conclusion drawn based on soil analysis (which relates to the underlying soil) would be correct only in absence of this deposition.

[Figure]

30) Page 23, line 16: The ITCZ is not expected to be convergence from the north, south and east at a given country. It is a large-scale structure, which changes in position every day, and it is closely linked to the Inter-Tropical front.

This sentence has been removed as a consequence to comment 4.

31) Page 23: lines 24-29: Transport and deposition of Saharan dust over the Sahel, which is afterward uplifted in a different season: This hypothesis is based on which scientific evidence? Are these speculations? If so, why they are mentioned?

The deposition of Saharan dust in Sahel is a well-known phenomenon. Once deposited, if the surface and wind conditions allow it, there would be no justification as to why this dust could not be re-suspended. As those wind and surface conditions change throughout the year in Sahel, it is indeed possible that dust is deposited during part of the year, and re-emitted during a different part of the year. The speculation here is not on the fact that there is dust deposition in Sahel or that after deposition dust could be re-suspended, but on the fact that it does happen instead of or in addition to suspension of dust from the underlying soil. Proving this hypothesis would require a specific study of its own. This discussion is removed from the revised manuscript.

32) Page 24: lines 13-16: "The conclusions of these two different analysis should probably be: : :" Here the authors of the manuscript justify a discrepancy with the conclusions from two other papers by telling that their data set should have been interpreted in a different way. This is strange and awkward for a scientific paper. One can tell conclusions from a published data set, but cannot change the conclusions from other scientists.

This is not what we meant. We did not question or interpret differently the two cited publications. We proposed that our analysis lead to different results because we did not use the ancillary data (winds in particular) in the same way as was done in the cited publications. The last sentence containing "the conclusions of these two different analyzes" referred to the two different ways to use ancillary data (and not to the two studies cited), therefore proposing a conclusion based on the combination of the already published studies together with the results from our analysis. However, the new way to account for winds and soil moisture now is closer to what was done in the cited publications (Marticorena 2010 and Kaly 2015) and leads to detecting plausible dust emissions in Sahel close to the 3 AMMA ground-based stations only during May to July, coherent with the cited results.

33) Page 25: line 9: "The Bodélé depression seems to be more active in the morning " where specifically is this shown in Figures 20 and 21? The Bodélé region (with marked limits) should be much better identify in the figures.

The Bodélé depression showed more surface dust events for local morning than for local evening (as many areas). In the new analysis using hourly wind values, the local evening occurrences of lowermost layer dust is much lower almost everywhere and the specific comment linked to Bodélé was removed. An interpretation was provided for this huge difference between local morning and local evening. (section 3.3 in the new

manuscript)

34) Maps are very noisy and it is difficult to know where red spots are in the same particular region from one to the other.

See answer to comment 2.

35) Page 25 line 14: "the situation might be different during winter" why this statement is not clear? The data sets shown in the paper do not show this?

The cited publication from Todd and Cavazos-Guerra only covers the summer months, as stated in the beginning of the sentence, line 13. Therefore, we don't know what that study would conclude for winter months, hence the use of the "might" word.

36) Page 25: lines 14-16: Why conclusions on a source region west of Bodélé are linked to those from Bodélé itself? This is statement is not clear and should be better explained.

The conclusion is based on the fact that those potential source areas were close, while they showed a different diurnal cycle. The most plausible reason for that is that they are linked to different emission mechanisms and different dust events. If they were showing similar diurnal cycles, it would be plausible either that they are both sources, or that low altitude transport occurs, making us detect dust in the lowermost layer at the source (Bodélé) and downwind. Now that the surface winds are considered on a hourly basis, this discussion is not anymore useful as the wind criterion by itself better separates local emission from low altitude transport.

37) Page 26: line 4: The sentence is not clear. Re-write it please.

Done. The whole paragraph was rewritten.

38) Page 26, lines 11-12: Consistency with LLJs during early spring is not clear. This should be explained in a much clear way.

That is not anymore discussed, as the morning versus evening comparisons are probably hampered by an underestimation of the 10m wind speed in ERA-5.

39) Section 3.5: The title of this section is not clear. It should be named "Inter-annual evolution" or similar. This section draws conclusions from a data set that is not shown. Evidence for these statements is not given. Therefore, either this section should be withdrawn or clear figures showing this inter-annual evolution should be presented.

This section was rewritten (now 3.4) and figures are now provided. No firm conclusions are drawn, only suggestions are made as the inter-annual variability is too high.

40) Page 27, line 21: "unique" means that only the MAPIR approach derives dust 3D data? This is not the case. Please correct.

Yes, this is what unique means. And yes it is the case. To our knowledge there is no other daily global dust 3D/profile data set over 9 years. There are climatologies from CALIOP, but those are climatologies, not day-to-day global data (impossible due to the low coverage by CALIOP). There exists another dust profile retrieval from IASI (Cuesta et al., 2015, doi:10.1002/2014JD022406) but only few data have been processed.

41) Page 27, line 27: "dust in deposition" is always occurring when close to the surface. There is not privileged place for this. Therefore, this cannot be specified as such.

The end of the sentence has been removed.

42) Page 27, line 30-32: The analysis of monthly average winds is not sufficient for this statement of transport from central Sahara.

End of sentence removed (from "and the wind patterns") consistently with the removal of that part of the analysis linked to comment 4.

43) Page 28, lines 1-4: Large dust emissions over the Sahel occur often by very strong winds (sporadic and possible missed by the filter) associated with Mesoscale Convective Systems.

The sentence in those lines is about dust south of the Sahel, not in the Sahel (we have modified "south of Sahel" to "at the south of the Sahel" to avoid confusion). Therefore in areas filtered out by the soil moisture and NDVI filters.

44) Page 28, line 16: "probability of local emission is high" how is this probability measured? It is quantified?

This sentence is modified now that the winds have been analyzed an a hourly basis.

45) Page 28, line 18: "global". This analysis is not global (worldwide). The term is not correct.

The paragraph was completely rewritten and this does not appear anymore.

46) Page 28, line 35: "good Earth coverage" this is not fully correct since it is not the case for single years as the example of 2015.

Answer to this comment is the same as to comment 3. The Earth coverage by the MAPIR data is good (almost all cloud-free scenes). The data with surface sensitivity was sparse in version 3.5, and is better in version 4.1.

**Answers to referee 2**

Major revisions

It is not common to have sub-sections in the Introduction. I would also suggest to shorten the Introduction to the relevant information. The details presented in Section 1.3 could be moved to Section 2 where instruments and methods are being described. Also, please provide the details on MAPIR in the main text rather than the Appendix. I found it confusing to read about results without any description of the used algorithm.

With the new version of MAPIR, a separate manuscript was prepared (algorithm and validation, Callewaert et al, AMTD 2019). Therefore in the revision of this manuscript the full description is not given anymore (the Appendix is removed) and a quick summary of the algorithm main points is provided in Section 2. Section 1.3 (added value of IASI) has been moved to the beginning of section 2 (after the added algorithm description). Section 1.1 (dust emission mechanisms) was moved upwards in the introduction, included in the main text before the climate effects of dust. Section 1.2 (short literature review of dust sources) was moved to the beginning of section 3 (results analysis). The introduction is therefore now in one section and much shorter.

I suggest to find a better name for what you call "surface layer" right now to avoid confusion with the actual surface layer. The lowermost kilometer or lowermost layer would do.

We have adopted "lowermost layer" as suggested.

From reading Section 2.1.1 it is not clear what quality control measures are being used. Overall, this section is not very focused and could be shortened.

In general, for any use of the MAPIR data, there is a standard quality control: the RMS of the difference between the observed spectrum and the simulated spectrum after retrieval (in the retrieval windows) must be lower than 2K in the old MAPIR v3.5 or 1K in the new MAPIR v4.1. This is the part that was explained together with the retrieval, in the Appendix. In the revised manuscript, a brief summary of the (new) algorithm is provided in the introduction and the full description and validation takes place in a different manuscript.

The section explaining this specific quality control was rewritten (section 2.1.4).

Section 2.1.2 also is not very focused and does not provide crucial information: Which region is being compared? What is meant with best comparison statistics? What is meant with dust detection for CALIOP and IASI. Is it only based on AOT or would CALIOP, for instance, need to show dust or polluted dust in the feature mask? The information in Figures 1, 2 and 4 would be much better presented in a Table that gives the percentage of the respective detection. Altogether, it is not clear what value is provided by the comparison between MAPIR and CALIOP surface dust detection if it is not quantitative. It is my impression that the information provided in this section could easily be conveyed in a paragraph. Figures 3, 5 and 6 should be omitted or moved to the supplementary material.

The compared region was the same as for the rest of the study, the whole North-Africa. The co-locations were plotted in Figures 3 and 5, showing the exact places of comparisons. "Best comparison statistics" meant that there were significant detections with IASI and CALIOP but that not all scenes were detected with dust (meaning a decent threshold for the dust detection). This was indeed a fully empirical approach, but it was explained so and one of the goals of this approach was to determine which aerosol concentration from MAPIR we could trust as being representative of the presence of dust in the lowermost layer. For CALIOP, of course we used only the data marked as dust or polluted dust, this was mentioned page 7 line 9.

In the revised manuscript, considering that now there is another manuscript with detailed validation, we decided to remove those attempts at comparing surface dust detections. This is also because actually close to sources, the time difference of 3 to 5 hours between IASI and CALIOP is not acceptable. Indeed, the atmospheric conditions could be completely different, with for example a dust emission event occurring at the IASI overpass, and completely finished at the CALIOP overpass time, leaving only the elevated dust layer. Comparisons are therefore always subject to assumptions, and bad comparisons are extremely difficult to interpret: are they due to bad results, or was

it a short event not seen by both instruments? In the algorithm description and valida-
tion manuscript, we used ground-based lidar data, and the CATS instrument on board
the International Space Station. Those allow for better time co-locations (we selected
maximum 1 hour), and therefore comparisons without the need of assumptions about
possible plume transport. Those comparisons show a very good capability of MAPIR
to detect dust plumes and reproduce their vertical extent, included in the lowermost
layer.

Section 2.2.1: Do you check in any way that the days of increased wind speed are the
same as the days with MAPIR dust detection in the lowermost layer? Could there be
a bias due to a mismatch of high-wind-speed days and dust days? Figure 7 seems
unnecessary as your criterion is fulfilled throughout most of the study region.

We have now moved to a more thorough analysis of the surface winds and soil mois-
ture. See answer to Major revision comment 5 from Referee 1 for the reason why we
had not done so before, and how it is done now.

Figures 7 to 12 should be omitted or moved to the Appendix or the Supplement as they
are hardly being referred to in the text, and thus, seem unnecessary. It would be better
to go straight to the findings for the full time series rather than focusing on a single
year.

Figures 7, 10 and 11 are not relevant in the revised manuscript where winds and soil
moisture are considered on an hourly and daily basis respectively. Figure 9 clearly
should also be removed especially if the other figures are removed. We think that figure
8 (the land cover mask) should be maintained, as this is a permanent filter feature with
no daily, seasonal or even yearly evolution.

At some point in the paper, it would be good to provide an overview map of known dust sources (could be combined with the information in Fig. 13) and a discussion on how dust sources are being defined in dust transport models. This would then allow to compare these source locations to your findings and point our possible impacts of your work.

The difficulty in providing an overview of the known dust sources is that depending on the reference one takes, there are variations. Plotting on a single map the whole literature on dust sources would be unrealistic and unreadable. A nice example of those differences is shown in Figure 3 of Schepanski 2012 (doi: 10.1016/j.rse.2012.03.019), showing on a single map their own results from 3 different satellite data sets. One may only imagine what would happen if plotting on a single map all the dust sources highlighted in all the previous dust studies. The discussion on the dust source definition in models is out of the scope of this manuscript.

I don't see the point of Section 3.1. The discussion doesn't seem to be within the scope of the paper. Also, not a single reference is being provided regarding circulation pattern over north Africa. I suggest to omit this section and the related Figures 15 to 17.

The basic discussion of the general transport patterns was also questioned by the other referee, and we agree that it was simplistic. As both referees had similar remarks regarding this, we decided to remove that small transport analysis attempt from the manuscript. Indeed, dust transport using our data would require in itself a dedicated study.

The discussion of dust detection either in the total column or in the the lowermost layer has its place in the manuscript. Using information in the lowermost layer is what makes this work new. Therefore a comparison with what we would have obtained with a similar study without the vertical resolution is totally in the scope of the paper.

Section 3.2 provides an overview of the dust sources. I am missing discussion of what makes those regions good dust sources. Also, a schematic map of the commonly known sources and new sources identified in this study would be good.

We do not understand what "good dust sources" mean. In this work, we identify dust sources as places where dust is detected close to the surface, together with sufficient surface wind and decent surface ancillary parameters (land cover, soil moisture, vegetation index). The request of a schematic map of commonly known dust sources is similar as the one for an overview of known dust sources and has been addressed.

Minor revisions

please avoid colloquial language such as good agreement or good representation.

Agreed and modified

page 2, line 1: I have heard of a semi-direct effect but not of semi-indirect ones.

After verification, this is a typing mistake: it should indeed have been semi-direct effects.

page 2, line 3: Do you mean the dust particles act as cloud condensation nuclei (CCN) and affect cloud droplets or as ice nucleating particles (INP) and affect cloud ice?

Dust particles can act both as Ice or Cloud Condensation Nuclei depending on the conditions. The sentence was indeed unclear and was rewritten.

[Figure]

page 2, lines 7-9: This paragraph could be omitted.

This paragraph shortly highlights the health and societal issues linked to dust and is therefore relevant in a general introduction of dust impacts, justifying the interest in studying dust sources as much as the climate effects do.

page 3, line 4: Replace first six words in line with Below

Done

page 4, line 13: change exploited to operated?

Done

page 14, lines 3-7: Please contact the authors to clarify instead of writing a speculative paragraph.

Done. During personal discussion with the first author (Sagar Parajuli), clarification was provided about that NDVI analysis and the content of the figure with respect to the text. The data between 0.18 and 0.28 NDVI are very noisy and does not show a decreasing trend. Therefore, we agree to our mistake here and we corrected to a threshold of 0.18 for NDVI. The effect of this change is limited to small areas in the Sahel, and is not seen in the comparison of the monthly maps but only in a difference map. So the change does absolutely not modify the general conclusions drawn from the analysis, although it does change slightly the numeric values of the surface dust detections after filtering.

page 22, line 18: What are FAO and IIASA?

FAO = Food and Agriculture Organisation of the United Nations

IIASA = International Institute for Applied Systems Analysis

As the soil analysis was removed from the manuscript, those acronyms do not figure in it anymore.

page 28, line 18: What is meant with a global study of that area?

The paragraph was completely rewritten.

page 28, line 23/24: You are contradicting your own work and it would be worthwhile to investigate the effect of not accounting for soil type in your analysis.

We are not contradicting our own work. The soil type was not used in the filters for dust source areas but only as additional discussion material, with the specific caution that it was unsure how relevant it is. However, as indeed it is unsure how relevant it is, and considering that it raised relevance questions in both the referees reviews, we decided to remove that part of the analysis.

Author contributions: From the description it seems to me that S. Vandenbussche should be the sole author of this paper. Supervision and reading a manuscript don't warrant co-authorship.

We strongly disagree. Supervisors have always been granted co-authorship for the specific work of their scientists. We would agree that the supervisor should not be granted co-authorship when in general the team's contribution to a larger manuscript

is small, but absolutely not in the case that the work is the product of the supervisor's scientist(s), done with his/her guidance.